# Distinct vaginal microbial signatures in pregnancies complicated by antiphospholipid syndrome: depletion of *Lactobacillus johnsonii* and enrichment of *Bifidobacterium dentium*

Yilin Fu,[1,2] Nary Long,[1,2] Phannaroat Sourn,[1,2] Weikun Li,[1,2] Zhenzhen He,[1,2] Weidong Tan,[1,2] Junjie Yuan,[1,2] Yuxin Chen,[1,2] Jianli Wu,[1,2] Shaoshuai Wang,[1,2] Ling Feng,[1,2] Zizhuo Wang,[1,2] Wencheng Ding[1,2]

**ABSTRACT** Antiphospholipid syndrome (APS) is a systemic autoimmune disease that contributes substantially to recurrent pregnancy loss, fetal death, intrauterine growth restriction, and preeclampsia, posing major threats to maternal and fetal health. These obstetric complications exhibit clinical similarities to those resulting from vaginal dysbiosis, yet the vaginal microbiota in APS pregnancies has not been systematically investigated. In this study, we characterized the vaginal microbiome in 33 pregnant women with APS and 90 healthy controls using 16S rRNA gene sequencing. We identified a unique microbial signature in APS pregnancies that differed from the commonly observed pattern of increased microbial diversity and *Lactobacillus* depletion seen in classical vaginal dysbiosis. Specifically, while overall alpha diversity and *Lactobacillus* dominance were preserved, we observed distinct compositional restructuring characterized by selective depletion of *Lactobacillus johnsonii* and marked enrichment of *Bifidobacterium dentium*. A logistic regression model integrating the relative abundances of these microbial biomarkers demonstrated robust diagnostic performance in differentiating pregnancies with APS from healthy pregnancies, with risk scores significantly correlating with clinical parameters and pregnancy outcomes. This study demonstrates that pregnant women with APS display a distinct vaginal microbiome pattern defined by species-specific compositional restructuring rather than global dysbiosis. These microbial alterations may contribute to APS-related pregnancy morbidity, highlighting vaginal microbial signatures as promising noninvasive biomarkers for risk stratification and potential therapeutic targets in obstetric APS management.

**IMPORTANCE** Antiphospholipid syndrome (APS) is an autoimmune disease that causes recurrent miscarriage, fetal death, and pregnancy complications in women of reproductive age. While coagulation dysfunction is a known contributing factor, whether APS is accompanied by vaginal microbiota alterations and their role in adverse outcomes remains unclear. We discovered that pregnant women with antiphospholipid syndrome harbor a unique vaginal microbial community: they exhibit depletion of the protective species *Lactobacillus johnsonii* while showing enrichment of *Bifidobacterium dentium*, a bacterium typically found in the gut. Unlike typical vaginal infections that display widespread microbial dysbiosis, antiphospholipid syndrome induces only selective alterations in specific bacterial species. These microbial signatures correlated with hematological parameters and adverse pregnancy histories, including prior miscarriages. Our findings suggest that monitoring vaginal microbiota could provide a simple, noninvasive approach to identify high-risk pregnancies in women with antiphospholipid syndrome and may guide novel screening strategies for pregnancy-related disorders targeting the vaginal microbiome.

**Peer Reviewer** Darina Cejkova, Vysoke uceni technicke v Brne, Brno, Czechia

Address correspondence to Wencheng Ding, dingwencheng326@163.com, Zizhuo Wang, wangzizhuo@tjh.tjmu.edu.cn, or Ling Feng, fltj007@163.com.

Yilin Fu, Nary Long, and Phannaroat Sourn contributed equally to this article. Author order was determined alphabetically.

The authors declare no conflict of interest.

See the funding table on p. 13.

KEYWORDS antiphospholipid antibodies, human microbiome, medical outcomes, autoimmunity

Antiphospholipid syndrome (APS) is a rare systemic autoimmune disease, with an estimated prevalence of approximately 0.05% in the general population, defined by the persistent presence of antiphospholipid antibodies (aPL), and clinically manifests with thrombosis and pregnancy morbidity (1, 2). Clinically, obstetric APS (OAPS) contributes substantially to recurrent pregnancy loss, fetal death, intrauterine growth restriction, and preeclampsia, posing major threats to maternal and fetal health (3). In clinical practice, APS is a leading cause of recurrent pregnancy loss, with studies indicating that APS-related pregnancy complications occur in approximately 15%–20% of women with a history of multiple miscarriages, often complicating the management of such patients (4, 5). Although thrombotic mechanisms play a central role, evidence increasingly supports nonthrombotic immune pathways in OAPS, including complement activation, neutrophil extracellular traps, and microRNA dysregulation (6, 7). These findings suggest that APS-related pregnancy complications involve complex immune-tissue interactions beyond classical coagulation disturbances.

Microbiota dysbiosis is closely linked to human disease. Emerging studies implicate the microbiome as a key modulator of systemic autoimmunity (8). Gut microbial dysbiosis has been linked to APS, rheumatoid arthritis, and other autoimmune and inflammatory disorders through mechanisms such as molecular mimicry and barrier disruption (9–11). For example, *Enterococcus gallinarum* can translocate across the gut mucosa and elicit autoimmunity in susceptible hosts (12). Similarly, the commensal *Roseburia intestinalis* has been shown to trigger cross-reactive responses in APS (13). The established role of microbiota in systemic autoimmunity raises the possibility that microbial ecosystems may influence development of autoimmune disorders, including OAPS.

Beyond the gut, the vaginal microbiome is another pivotal ecosystem, representing a critical interface between mucosal immunity and pregnancy outcomes. In reproductive-age individuals, the vaginal microbiome is often categorized into five community state types based on the dominant bacterial species (14). During pregnancy, the microbiome generally stabilizes toward a *Lactobacillus*-dominated state. A *Lactobacillus crispatus*-dominated microbiome is particularly associated with health due to its role in maintaining acidic pH, inhibiting pathogens, and regulating local inflammation (15). Conversely, dysbiosis characterized by reduced *lactobacilli* and overgrowth of anaerobes is associated with adverse outcomes such as preterm birth, preeclampsia, and recurrent miscarriage (16–18). These conditions overlap with OAPS manifestations.

Despite this overlap, the vaginal microbiota in APS pregnancies has not been systematically investigated. We hypothesized that pregnant women with APS harbor a distinct vaginal microbiota composition; understanding whether OAPS is accompanied by unique vaginal microbial patterns could reveal previously unrecognized pathogenic pathways and potential diagnostic or therapeutic targets.

Therefore, this study aimed to characterize the vaginal microbiome composition in pregnant women with APS, identify key microbial alterations, and evaluate their associations with clinical parameters and adverse pregnancy outcomes.

## MATERIALS AND METHODS

### Study design and population

This case-control study was conducted at Tongji Hospital, Tongji Medical College, Huazhong University of Science and Technology (a major tertiary referral center) from March 2024 to March 2025. We aimed to recruit the APS group and controls (healthy pregnant women without APS) matched for gestational age at sampling. Eligible participants were women with singleton pregnancies, recruited in the third trimester during routine screening. Exclusion criteria were multiple gestations, hypertensive

disorders of pregnancy, gestational diabetes mellitus, thyroid dysfunction, hepatitis B infection, Group B *Streptococcus* colonization, vaginal candidiasis, systemic lupus erythematosus, connective tissue diseases, use of antibiotics or antifungals within 2 weeks, and sexual activity or vaginal douching within 72 h prior to sampling. All patients in the APS group were diagnosed according to the revised Sapporo (Sydney) classification criteria (19). Diagnosis required the fulfillment of at least one clinical criterion (including recurrent unexplained miscarriage before 10 gestational weeks or unexplained fetal death at ≥10 gestational weeks) and one laboratory criterion, defined as persistent positivity for lupus anticoagulant, anticardiolipin antibodies (IgG/IgM), or anti-$\beta_2$-glycoprotein I antibodies (IgG/IgM) on at least two occasions ≥12 weeks apart. All diagnoses were confirmed by rheumatologists or maternal-fetal medicine specialists. Among the APS group, some patients had been diagnosed with APS within 2 years prior to pregnancy, while others were newly identified during the current pregnancy; however, all cases fulfilled the revised Sydney classification criteria. All participants were under the care of two senior obstetricians with more than 20 years of clinical experience and received standard pharmacological management (20).

Clinical data were retrospectively collected. Maternal characteristics included age, gravidity, and parity. Pregnancy and delivery outcomes comprised gestational age at delivery, mode of delivery (categorized as cesarean section or vaginal delivery), and the occurrence of specific complications: preterm birth (<37 weeks), small for gestational age (birth weight <10th percentile), large for gestational age (birth weight >90th percentile), premature rupture of membranes, and placental abruption. Neonatal outcomes included neonatal birth weight, birth weight percentile (calculated using the INTERGROWTH-21st standards), Apgar scores at 1 and 5 min, and admission to the neonatal intensive care unit (NICU). Maternal laboratory indices included complete blood count indices, including red blood cell (RBC) count, platelet count (PLT), and coagulation parameters, including antithrombin activity, activated partial thromboplastin time, and thrombin time.

## Vaginal sample collection and DNA extraction

All vaginal swab specimens were collected by two well-trained operators following the standard operating procedure. A sterile, single-use flocked swab (BKMAN Biotech Co., Ltd) was gently inserted into the posterior vaginal fornix under direct visualization with a speculum, avoiding any contact with the external genitalia. The swab was then gently rotated 10 times, immediately placed into a sterile cryovial, and stored at −80℃. After sample collection, microbial genomic DNA was extracted from all vaginal swabs using the FastPure Stool DNA Isolation Kit (MJYH, Shanghai). DNA concentration and purity were determined using a NanoDrop 2000 spectrophotometer (Thermo Scientific). DNA integrity was checked by 1% agarose gel electrophoresis.

## DNA amplification

The V1-V9 hypervariable regions of the 16S rRNA gene were amplified by PCR using barcoded primers 27F (5′-AGRGTTYGATYMTGGCTCAG-3′) and 1492R (5′-RGYTACCTTGT-TACGACTT-3′). Each 20 µL PCR reaction contained 4 µL of 5× FastPfu buffer, 2 µL of 2.5 mM dNTPs, 0.8 µL each of forward and reverse primers (5 µM), 0.4 µL of FastPfu DNA polymerase, 0.2 µL of BSA, and 10 ng of template DNA. Reactions were performed in triplicate for each sample. The PCR amplification was performed as follows: initial denaturation at 95 ℃ for 3 min; 27 cycles of denaturation at 95 ℃ for 30 s, annealing at 60 ℃ for 30 s, and extension at 72 ℃ for 30 s; followed by a final extension at 72 ℃ for 10 min, and storage at 4 ℃ (T100 Thermal Cycler PCR thermocycler, Bio-Rad, USA). PCR products were verified by 2% agarose gel electrophoresis, purified using the AMPure PB beads (Pacific Biosciences, CA, USA), and quantified with Qubit 4.0 (Thermo Fisher Scientific, USA).

## DNA library construction and sequencing

Purified amplicons were pooled in equimolar concentrations, and DNA libraries were constructed using the SMRTbell Prep Kit 3.0 (Pacific Biosciences, CA, USA) following the manufacturer's protocol. The purified SMRTbell libraries were sequenced on the PacBio Sequel IIe platform (Pacific Biosciences, CA, USA) by Majorbio Bio-Pharm Technology Co., Ltd. (Shanghai, China). High-fidelity (HiFi) reads were generated from the subreads using circular consensus sequencing implemented in SMRT Link v.11.0 and were demultiplexed using barcode sequences, filtered for length (1,000–1,800 bp). To ensure sequence quality, chimeric sequences were identified and removed using the DADA2 algorithm. The optimized HiFi reads were subsequently clustered into operational taxonomic units (OTUs) at a 97% sequence similarity threshold using UPARSE version 11. The most abundant sequence for each OTU was selected as a representative sequence. Chloroplast-derived sequences were removed from the OTU table. All samples were rarefied to a uniform depth of 20,368 16S rRNA gene sequences.

## Bioinformatic analysis

Bioinformatic analysis was performed using the Majorbio Cloud platform (https://cloud.majorbio.com). Downstream statistical analyses were conducted based on the rarefied OTU table. The taxonomic classification of each OTU representative sequence was carried out using the RDP Classifier (version 2.2) against the 16S rRNA gene database (NT_Taxon_core_v2024) with a confidence threshold of 0.70. Alpha diversity was assessed using the Shannon index calculated with Mothur v.1.30.1. Beta diversity was determined based on both Bray-Curtis and Canberra dissimilarity metrics, visualized via principal coordinate analysis (PCoA) and non-metric multidimensional scaling (NMDS). The statistical significance of group separation in beta diversity was tested using analysis of similarities. Differences in microbial community composition between groups were assessed using Wilcoxon rank-sum tests. The linear discriminant analysis (LDA) effect size (LEfSe) was performed to identify the significantly abundant taxa (phylum to species) of bacteria among the different groups (LDA score > 2, $P < 0.05$).

## Statistical analysis

Statistical analyses were conducted using IBM SPSS Statistics for Windows, Version 27.0 (IBM Corp., Armonk, NY). The normality of all continuous variables was assessed using the Shapiro-Wilk test. Based on this assessment, continuous variables with normal distribution were analyzed using two-tailed Student's $t$-tests, while non-normally distributed data were assessed with Wilcoxon rank-sum tests. Categorical variables were compared with the $\chi^2$ test or Fisher's exact test as appropriate. $P < 0.05$ was considered statistically significant. Spearman's rank correlation analysis was conducted to assess relationships between vaginal microbiota and clinical parameters. Multiple testing was adjusted using the Benjamini-Hochberg false discovery rate (FDR) method within each analysis tier, with q < 0.05 considered significant. GraphPad Prism version 10.0.0 (GraphPad Software, Boston, Massachusetts, USA; https://www.graphpad.com/) was employed for the generation of receiver operating characteristic (ROC) curves and scatter plots. Internal validation of the ROC model was performed in R software (version 4.5.1) using fivefold cross-validation repeated 100 times.

## RESULTS

### Participants and clinical characteristics

The recruitment flowchart is presented in Fig. 1. Among 421 pregnant women initially invited to participate, 281 consented and provided valid vaginal swab samples. A total of 158 women were excluded according to predefined exclusion criteria (mainly metabolic/endocrine disorders, other autoimmune diseases, infectious conditions, and sampling outside the gestational window). The final cohort comprised 123 participants, including 33 pregnant women meeting the revised Sydney criteria for APS and 90

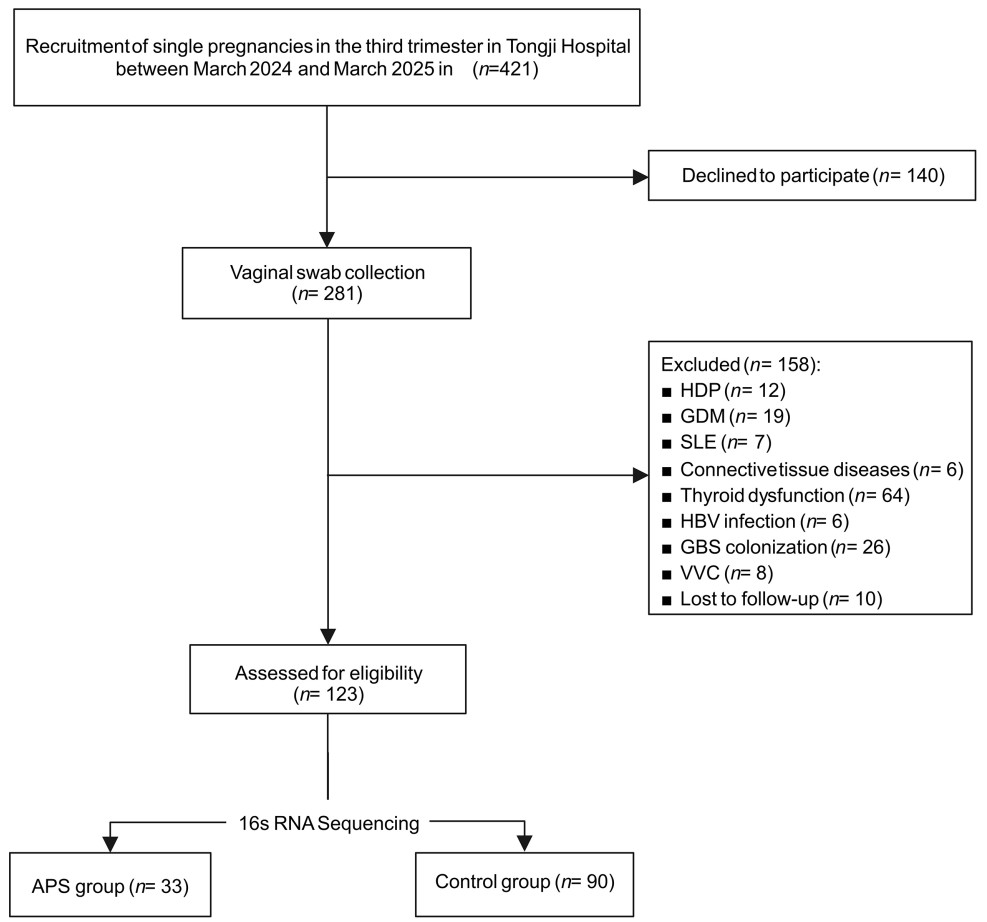

**FIG 1** Participant recruitment flowchart. A total of 421 pregnant women were invited, 281 consented and underwent vaginal swab collection. After exclusion based on predefined criteria, 123 participants remained, comprising 33 with APS and 90 healthy controls. APS, antiphospholipid syndrome; HDP, hypertensive disorders of pregnancy; GDM, gestational diabetes mellitus; SLE, systemic lupus erythematosus; GBS, group B *Streptococcus*; VVC, vulvovaginal candidiasis; HBV, hepatitis B virus.

matched healthy pregnant women. Of the 33 APS pregnancies, 19 had been diagnosed prior to pregnancy (Pre-APS), and 14 were newly diagnosed during the index pregnancy (Index-APS). Baseline clinical characteristics are summarized in Table 1. The two groups were matched for gestational age at sampling, as per design. Compared with controls, the APS group was slightly older than the control group (median age: 33.0 vs 31.0 years, $P = 0.024$), and had higher gravidity (2.73 ± 1.28 vs 1.50 ± 0.78, $P < 0.001$), more frequent spontaneous abortion (1.52 ± 1.09 vs 0.38 ± 0.63, $P < 0.001$), and a greater proportion of cesarean delivery (87.9% vs 56.7%, $P = 0.001$). In terms of pregnancy outcomes, the APS group had a lower birth weight percentile (29.8% vs 45.8%, $P = 0.039$) and a higher incidence of placental abruption (12.1% vs 2.2%, $P = 0.044$). Notably, comparisons of microbial diversity and community composition between the Pre-APS subgroup and the Index-APS subgroup revealed no significant differences (Fig. S1A through C).

## Vaginal microbial diversity

Rarefaction and pan-core analyses based on OTU-level confirmed adequate sequencing depth and comprehensive coverage of vaginal microbiome diversity within the study population (Fig. S2A through C). Alpha diversity analysis (Shannon index) showed no statistically significant differences (all $P > 0.05$) between the APS and control groups (Fig. 2A), suggesting the two groups had similar overall levels of species richness and evenness.

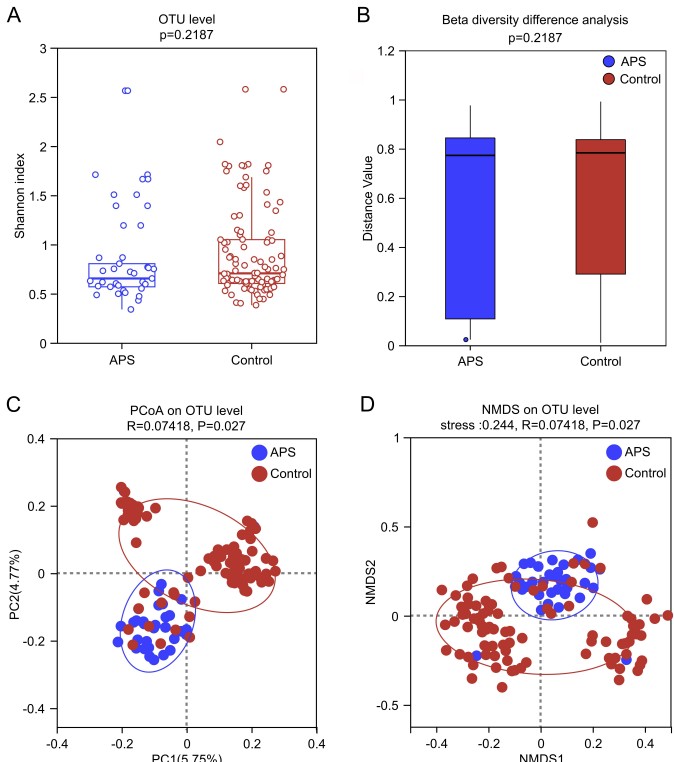

**FIG 2** Vaginal microbial diversity in APS and control groups. (A) Alpha diversity, determined using the Shannon index. (B) Beta diversity analysis based on OTU level is assessed using Bray-Curtis distances. (C, D) Beta diversity analysis at the OTU level was assessed using Canberra distancesas, as illustrated in (C) PCoA plots and (D) NMDS.

Using the Bray-Curtis distance, which is sensitive to overall community variation, we calculated the within-group dissimilarities for each group (Fig. 2B). The APS group exhibited a trend of lower within-group heterogeneity (0.77, IQR: 0.11–0.84) compared to the control group (0.78, IQR: 0.29–0.84), although this difference did not reach the significance threshold ($P = 0.087$). This finding is consistent with the results from the Venn diagram, which showed that the APS group harbored fewer unique taxa at various taxonomic levels than the control group (Fig. S3A through D). This suggests that the microbial composition in the APS group may be more homogeneous and possibly associated with a more consistent internal community state.

To further investigate subtle differences in the overall community structure between the two groups, we conducted an analysis using the Canberra distance, which is more sensitive to changes in rare taxa. NMDS and PCoA of OTU abundances revealed a moderate level of ordination fit and demonstrated statistically significant effects of group on beta diversity ($R = 0.074$, $P = 0.027$) (Fig. 2C and D). These findings suggest that while overall microbial diversity remains comparable between groups, APS pregnancies may harbor distinct microbial community structures.

## Clustering analysis reveals a disease-associated vaginal microbiome type

To further decipher the community structure captured by the Canberra distance, we performed hierarchical clustering on the species-level abundance data. This analysis delineated three primary clusters, which we designated Canberra Type 1 (CT1), CT2, and CT3 (Fig. 3A). Strikingly, the APS group samples were almost exclusively concentrated

**TABLE 1** The characteristics and pregnancy outcomes of APS and control groups[d,e]

| Characteristics/Outcomes | Control group (n = 90) | APS group (n = 33) | P-value |
|---|---|---|---|
| Maternal ages (years) | 31.0 (29.0–33.3) | 33.0 (30.0–35.0) | 0.024[c] |
| Gravidity | 1.50 (± 0.78) | 2.73 (± 1.28) | <0.001[b] |
| Parity | 1.12 (± 0.36) | 1.22 (± 0.42) | 0.067[b] |
| Spontaneous abortion | 0.38 (± 0.63) | 1.52 (± 1.09) | <0.001[b] |
| Gestational ages at sampling (week) | 37.0 (36.4–37.9) | 38.0 (36.8–38.6) | 0.024[c] |
| Gestational ages at delivery (week) | 39.40 (± 0.85) | 38.41 (± 1.01) | <0.001[c] |
| Delivery mode (%) | | | |
| Cesarean section | 51 (56.7) | 29 (87.9) | 0.001[a] |
| Vaginal delivery | 39 (43.3) | 4 (12.1) | |
| Apgar score at 1 min | 8 (± 0.00) | 8 (± 0.00) | 0.999[b] |
| Apgar score at 5 min | 9 (± 0.00) | 9 (± 0.00) | 0.999[b] |
| Birth weight (g) | 3,170 (2,983–3,413) | 3,070 (2,840–3,225) | 0.068[c] |
| Birth weight percentile(%) | 45.8 (24.9–68.5) | 29.8 (16.7–53.7) | 0.039[c] |
| Preterm birth | 1 (1.1) | 2 (6.1) | 0.175[a] |
| NICU admission | | | |
| SGA | 7 (7.8) | 5 (15.2) | 0.302[a] |
| LGA | 10 (11.1) | 3 (9.1) | 1.000[a] |
| PROM | 11 (12.2) | 4 (12.1) | 1.000[a] |
| Placental abruption | 2 (2.2) | 4 (12.1) | 0.044[a] |

[a]Chi-square test.
[b]Student's t-tests.
[c]Wilcoxon rank-sum test.
[d]Values for continuous variables are presented as means ± SD or medians (IQR); values for categorical variables are expressed as n (%).
[e]SGA, small for gestational age; LGA, large for gestational age; PROM, premature rupture of membranes.

within CT2 (31 of 33 APS samples), while healthy controls were distributed across CT1 and CT3. This indicates a high degree of microbial homogeneity among APS pregnancies that is distinct from the variation observed in healthy pregnancies. Visualization of the taxonomic composition (Fig. 3B) and identification of the most differentially abundant species (Fig. 3C) within these clusters confirmed that CT2 was characterized by the unique vaginal microbiome pattern: significant enrichment of *Bifidobacterium dentium* and depletion of *Lactobacillus johnsonii*.

## Vaginal microbiome differences between APS and control groups

At the genus level, *Lactobacillus* dominated the vaginal microbiome composition in both groups. *Bifidobacterium* was more abundant in the APS group than in the control group, although this difference did not reach statistical significance (8.25% vs 1.01%; $P = 0.055$) (Fig. 4A). The relative abundances of *Enterococcus* (0.39% vs 0.10%; $P < 0.001$) and *Campylobacter* (0.312% vs 0.064%; $P = 0.006$) were significantly higher in the APS group (Fig. 4B).

At the species level, *Bifidobacterium dentium* was markedly enriched in the APS group (7.188% vs 0.153%; $P = 0.017$), whereas *Lactobacillus johnsonii* was significantly depleted (0.671% vs 7.753%; $P < 0.001$) (Fig. 4C and D). In addition, *L. delbrueckii* (0.775% vs 0.045%; $P = 0.033$), *Enterococcus faecalis* (0.386% vs 0.074%; $P < 0.001$), and *Campylobacter ureolyticus* (0.310% vs 0.063%; $P = 0.015$) were significantly enriched in the APS group, while *Fannyhessea vaginae* (0.019% vs 0.895%; $P < 0.001$) was notably reduced compared with controls.

LEfSe analysis (LDA > 2.0, $P < 0.05$) further revealed distinct microbial signatures between the two groups (Fig. 4E). Microorganisms enriched in the APS group included *B. dentium*, *Enterococcus*, and *C. ureolyticus*, whereas *L. johnsonii*, *Gardnerella vaginalis*, and *F. vaginae* were predominant in the control group. Collectively, these findings indicate that pregnant women with APS exhibit a characteristic vaginal microbiome pattern

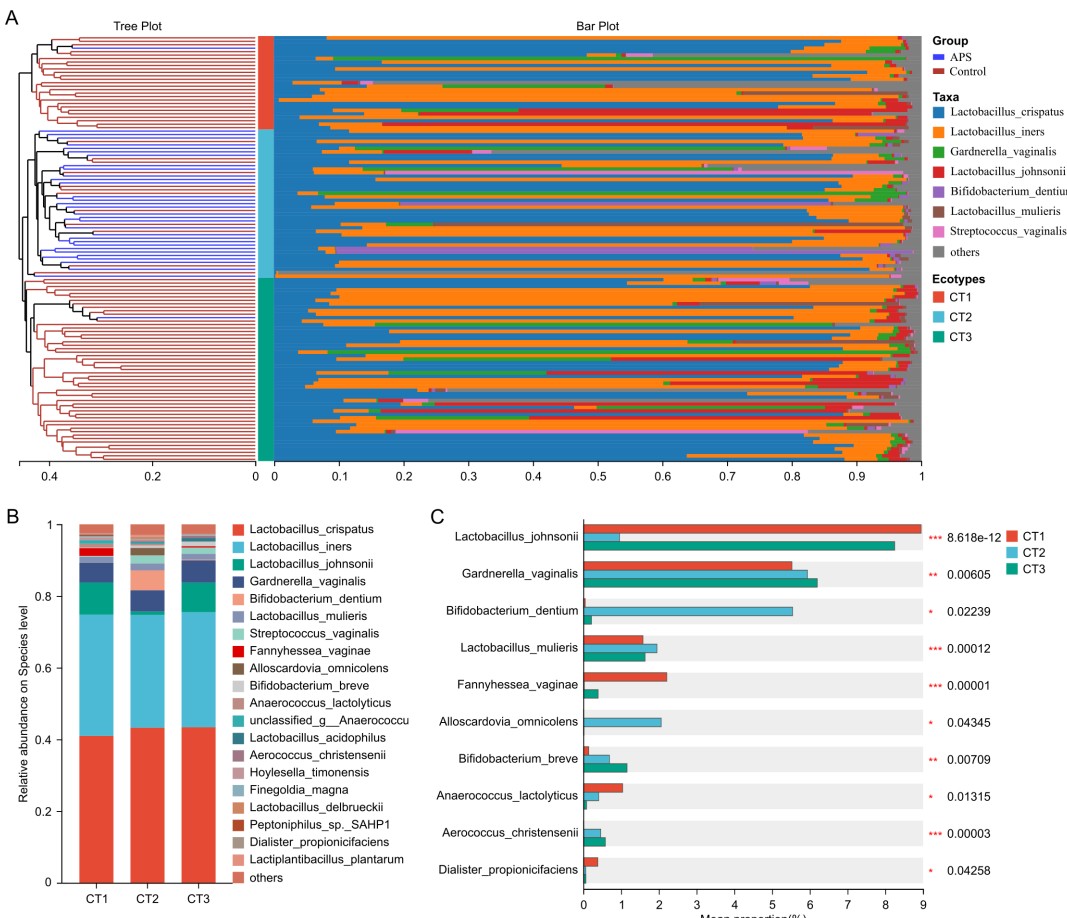

**FIG 3** Vaginal microbiota community structure across ecotypes. (A) Hierarchical clustering of individual vaginal microbiota profiles with corresponding stacked bar plots showing the relative abundance of dominant taxa; sample group (APS vs control) and ecotype classification (CT1–CT3) are indicated. (B) Mean relative abundance of bacterial species across ecotypes (CT1–CT3). (C) Differentially abundant taxa among ecotypes; *P*-values are shown (red asterisks indicate statistical significance).

defined by *L. johnsonii* depletion and *B. dentium* colonization rather than global microbial dysbiosis.

## Association between vaginal microbiome and clinical characteristics

Exploratory Spearman correlation analysis was performed to assess associations between vaginal taxa and clinical parameters. After applying FDR correction for multiple comparisons, none of the correlations remained statistically significant (all q > 0.05). The patterns of nominal (uncorrected) associations are displayed in a heatmap (Fig. 5A). Regarding hematological parameters, *B. dentium* abundance showed nominal negative correlations with both RBC count ($r = -0.220$, $P = 0.016$) and PLT ($r = -0.190$, $P = 0.038$), whereas *L. johnsonii* abundance showed a nominal positive correlation with RBC count ($r = 0.237$, $P = 0.009$). In terms of pregnancy-related variables, *B. dentium* showed a nominal positive correlation with gravidity ($r = 0.226$, $P = 0.012$) and spontaneous abortion history ($r = 0.225$, $P = 0.012$), but a nominal negative correlation with gestational age at delivery ($r = -0.238$, $P = 0.010$). *L. johnsonii* displayed opposing nominal association patterns with these same variables. These exploratory findings suggest potential links between the vaginal microbiome and clinical features in APS that warrant validation in larger, dedicated cohorts.

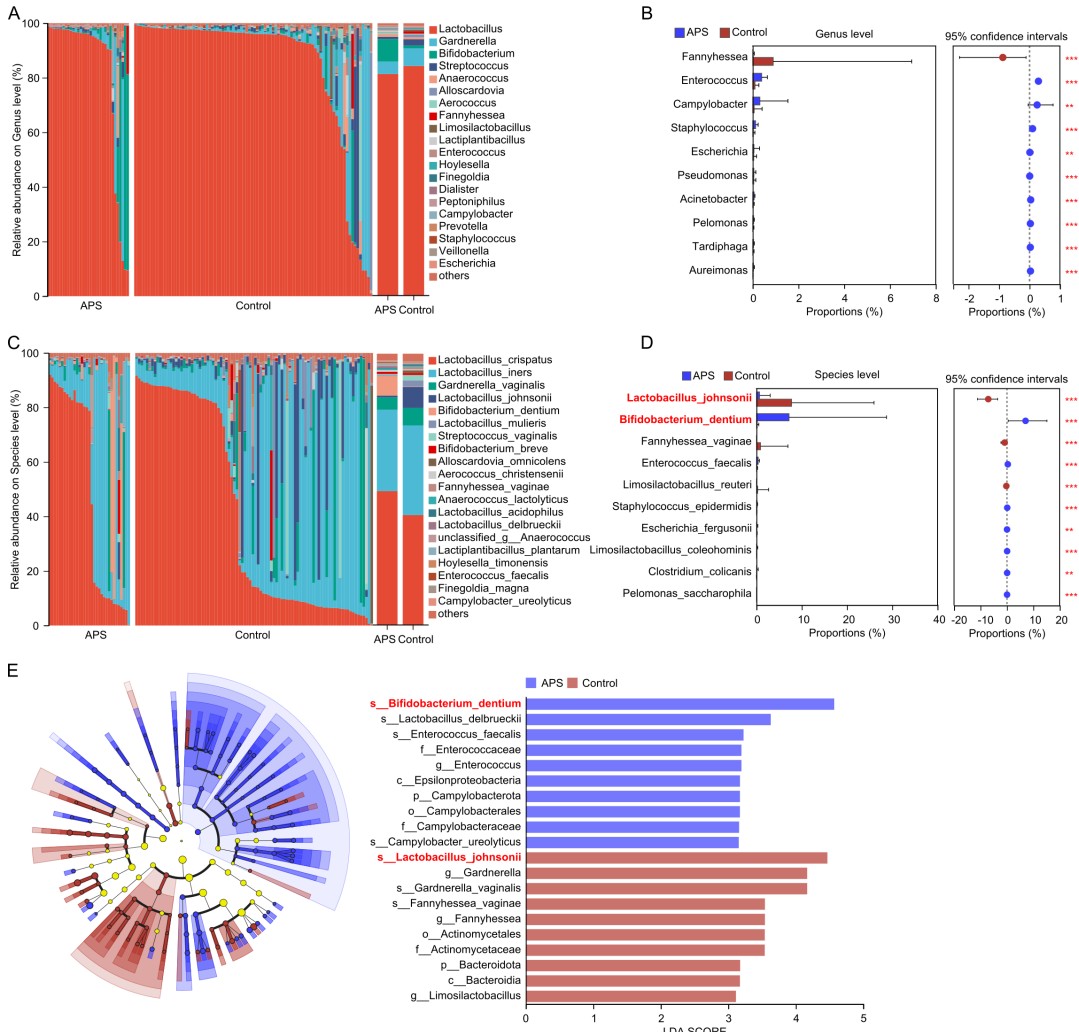

**FIG 4** Vaginal microbiome differences between APS and control groups. (A) Bar plots showing the relative abundance of dominant genera. (B) Differential genus-level abundance analysis comparing the APS and control groups, using the Wilcoxon rank-sum test. (C) Bar plots showing the relative abundance of dominant species. (D) Differential species-level abundance analysis comparing the two groups, using the Wilcoxon rank-sum test. (E) LEfSe analysis illustrating taxa enriched in each group, with the LDA score histogram highlighting discriminatory microbial taxa.

## Microbial biomarker-based risk scoring model

We next developed a logistic regression-based risk score using species-level relative abundances of key microbial biomarkers. The multivariate model integrating both microbial biomarkers achieved an AUC of 0.824 (95% CI, 0.746–0.902; $P < 0.001$), with sensitivity 81.8% and specificity 77.5% at the optimal cutoff, indicating robust diagnostic performance in discriminating APS from healthy pregnancies (Fig. 6A). An internally validated ROC curve further confirmed the robustness of this diagnostic model, yielding an AUC of 0.822 (95% CI, 0.813–0.830) with sensitivity 77.1% and specificity 73.1% at the optimal cutoff (Fig. 6B). Moreover, risk scores were negatively correlated with RBC count ($r = -0.240$, $P = 0.009$), gravidity ($r = -0.248$, $P = 0.007$), and gestational age at delivery ($r = -0.267$, $P = 0.004$) (Fig. S4C and D), but positively correlated with spontaneous abortion history ($r = 0.283$, $P = 0.009$) (Fig. 6C and D). Collectively, higher risk scores were associated with hematological abnormalities and adverse pregnancy histories, supporting the potential clinical utility of microbiome-based models for enabling individualized risk stratification.

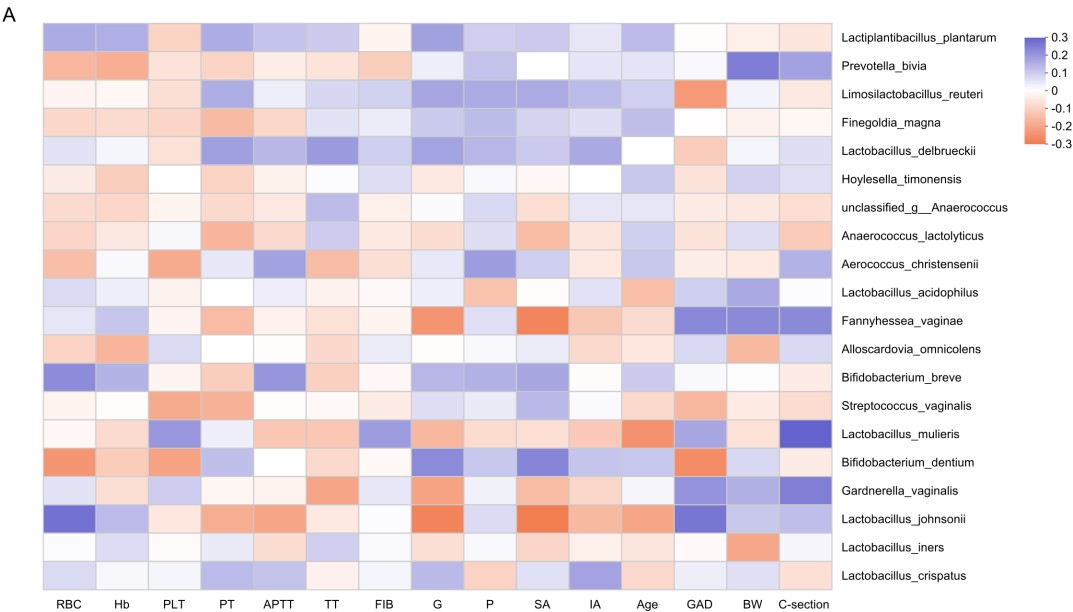

**FIG 5** Associations between the vaginal microbiome and clinical characteristics. (A) Heatmap showing correlations between vaginal microbial species and clinical parameters, calculated using Spearman's rank correlation. *P*-values were adjusted for multiple comparisons using the Benjamini-Hochberg FDR procedure. Asterisks within cells denote the significance level of the FDR-adjusted q-value: *q < 0.05, **q < 0.01, ***q < 0.001. The color gradient represents the strength and direction of correlation, with red indicating positive correlations and blue indicating negative correlations. Abbreviations: RBC, red blood cell; Hb, hemoglobin; PLT, platelet count; PT, prothrombin time; APTT, activated partial thromboplastin time; TT, thrombin time; FIB, fibrinogen; G, gravidity; P, parity; SA, spontaneous abortion; IA, induced abortion; GAD, gestational age at delivery; BW, birth weight; C-section, cesarean section.

## DISCUSSION

To our knowledge, this study provides the first comprehensive characterization of the vaginal microbiome in pregnant women with APS. Distinct from classical pregnancy complications like preterm birth and preterm premature rupture of membranes, which are typically marked by reduced *Lactobacillus* abundance and increased diversity (16, 17, 21), our findings reveal a unique microbial signature in APS pregnancies. Specifically, we observed preserved overall *Lactobacillus* dominance and stable alpha diversity, but with a distinct compositional restructuring marked by the depletion of *Lactobacillus johnsonii* and enrichment of *Bifidobacterium dentium*.

This species-specific dysbiosis suggests that APS is associated with a selective reshaping of the microbial ecology, rather than inducing a global microbial dysbiosis. Similar observations have been reported in pregnancy complicated by type 1 diabetes, where immune dysregulation shapes microbial community structure without reducing overall diversity (22). These parallels highlight that distinct maternal disease contexts may be associated with unique microbial signatures.

Our unsupervised clustering analysis not only identified a distinct and highly homogeneous vaginal microbiome state (CT2) encompassing the majority of pregnancies complicated by APS, but also revealed that the healthy vaginal microbiome in late gestation could be further stratified into at least two additional community types (CT1 and CT3), primarily distinguished by variation in low-abundance taxa. Although the ecological drivers and structural characteristics underlying this substructure within the healthy state remain to be elucidated and were beyond the scope of the present study, its presence nonetheless supports the non-random nature of the tight clustering of APS samples into a CT2 vaginal community characterized by enrichment of *Bifidobacterium dentium* and depletion of *Lactobacillus johnsonii*. Collectively, these findings suggest that this microbiome configuration is strongly linked to and may be a consequence of APS-specific pathophysiological mechanisms.

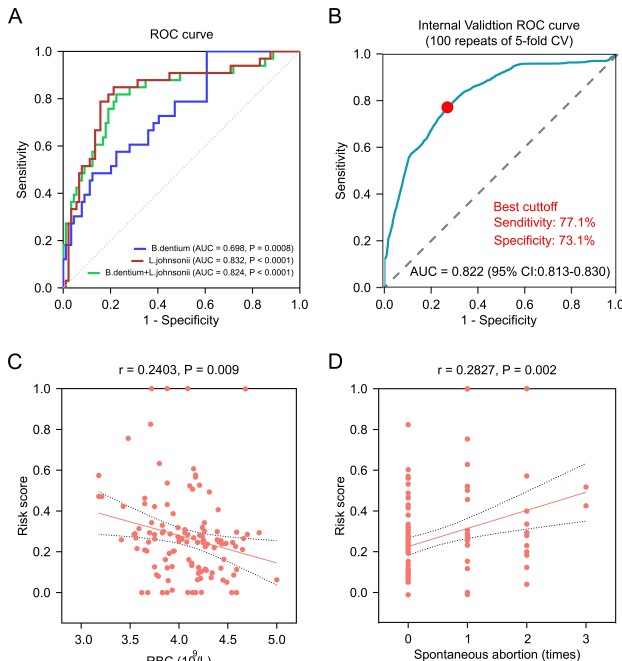

**FIG 6** Microbial biomarker-based risk scoring model. (A) ROC curves showing the diagnostic performance of *Bifidobacterium dentium*, *Lactobacillus johnsonii*, and their combination for distinguishing APS from controls. (B) Internal validation of the ROC model was performed using fivefold cross-validation repeated 100 times. (C) Scatter plots showing correlations between risk scores and RBC count, assessed using Spearman's correlation. (D) Scatter plots showing correlations between risk scores and spontaneous abortion, assessed using Spearman's correlation.

Beyond descriptive microbiome profiling, our study further established a logistic regression-based risk-scoring model integrating the relative abundances of *L. johnsonii* and *B. dentium*. The model demonstrated robust diagnostic accuracy and clinically meaningful correlations with hematologic and adverse pregnancy histories—features consistent with APS-related morbidity. Compared with conventional APS diagnosis, which relies on antiphospholipid antibody (aPL) testing and often requires repeated serologic confirmation over 12 weeks (20), microbiome-based screening could provide a complementary, rapid, and non-invasive diagnostic adjunct. This is exemplified by the work of Zheng et al., who developed a gut microbiota-based model for inflammatory bowel disease that provides high sensitivity and clinical cost-effectiveness (23). Vaginal microbial signatures can be detected directly from swab samples without venipuncture, offering potential for community-level risk stratification.

The selective loss of *L. johnsonii* is particularly noteworthy. As one of the protective *Lactobacillus* species in the vaginal microbiome, it maintains a pathogen-hostile acidic milieu, reinforces epithelial barrier integrity, and also promotes immune tolerance, while *L. johnsonii* additionally suppresses NF-κB–mediated inflammation, lowers pathogen burden, and boosts IL-10 to counter *Gardnerella*-induced dysbiosis (24, 25). Its depletion is thus positioned as a potential factor that may reduce microbial resilience and increase susceptibility to dysbiosis, particularly within the broader context of systemic immune dysregulation characteristic of autoimmune diseases such as APS, where widespread mucosal microecological imbalance including the oral, gut, and vaginal microbiota has been increasingly recognized (26–29). APS-associated immune activation driven by antiphospholipid antibodies, complement activation, and endothelial dysfunction may disrupt epithelial-immune crosstalk and alter mucosal environments (30, 31). Consequently, this immune-mediated instability may be permissive for the colonization of atypical or opportunistic taxa, such as *B. dentium*.

The enrichment of *B. dentium* is intriguing given that it is typically an intestinal commensal associated with mucosal metabolism and immune modulation (32). However, under certain conditions, it can exhibit pathogenic potential, as observed in dental caries and mucosal inflammation (33, 34). The detection of *B. dentium* in the vaginal tract—where it is not normally resident—suggests aberrant translocation or ecological displacement. This organism's tolerance to acidic environments and capacity to produce short-chain fatty acids and γ-aminobutyric acid may modulate local immune signaling and alter epithelial function (35, 36). Within the APS setting, such metabolic activity could disrupt the epithelial-immune equilibrium of the vaginal niche, further aggravating the consequences of *L. johnsonii* depletion.

Considering that APS pathophysiology compromises endothelial and mucosal barrier function (13, 30), hematogenous or ascending translocation from other mucosal sites could facilitate vaginal colonization by *B. dentium*. Such translocation phenomena have been reported in systemic diseases where gut microbes migrate to extraintestinal niches and contribute to tissue-specific inflammation (37–40). Moreover, pregnancy itself is known to elevate salivary *B. dentium* abundance (41), and the organism's tolerance to acidic environments and modest nutrient requirements may further facilitate its survival in the vaginal tract (36). These findings collectively support a plausible mechanism in which APS-associated immune dysfunction and barrier impairment enable selective microbial translocation and niche adaptation within the reproductive tract.

Conceptually, this study expands the paradigm of APS from a purely vascular and immunologic disorder to one that encompasses host-microbe interactions within mucosal ecosystems. It highlights that APS-associated pregnancy complications may be linked not only to thrombosis or complement activation but also to specific alterations in the vaginal microbiota, such as the depletion of *Lactobacillus johnsonii* and enrichment of *Bifidobacterium dentium*. This perspective opens new research directions exploring microbiome-immune interaction as a contributing mechanism to obstetric morbidity in APS. Clinically, our risk-scoring model based on *L. johnsonii* and *B. dentium* abundance showed promising predictive performance. If validated in larger, multi-center cohorts, vaginal microbial signatures could serve as rapid, cost-effective biomarkers for risk stratification in APS pregnancies.

## Strengths and limitations

The strengths of this study lie in its novelty, rigorous design, and translational focus. It represents the first systematic analysis of the vaginal microbiome in APS pregnancies, filling a major gap in reproductive immunology. The strict inclusion and exclusion criteria minimized confounding factors such as infection, antibiotic use, and metabolic disease. Finally, the development of a validated microbial risk score for APS pregnancies provides a foundation for integrating microbiome profiling into risk assessment frameworks.

Several limitations warrant acknowledgment. First, as a descriptive and exploratory study, the cross-sectional nature and limited sample size, particularly within the APS subgroup, precludes robust assessment of associations between specific microbial alterations and individual pregnancy complications and precludes causal inference. Prospective, longitudinal investigations tracking vaginal microbiome dynamics from preconception through postpartum are required to clarify temporal relationships. Second, the absence of non-pregnant women with APS as controls limits our ability to determine whether the observed microbiome differences are due to APS itself or to pregnancy-related effects. Further studies are therefore needed to characterize the vaginal microbiome in non-pregnant APS patients. Furthermore, this is a single-center study in which all participants were recruited from the same hospital over a one-year period. Finally, the use of 16S rRNA gene sequencing, while providing taxonomic resolution, precludes functional inferences. Therefore, the generalizability and broader applicability of the findings need to be validated in future multicenter, large-sample prospective cohort studies, ideally incorporating multi-omics approaches such as metagenomics and metabolomics to elucidate functional mechanisms. Nonetheless,

this study provides the first systematic description of vaginal microbiome alterations in pregnant women with APS, offering important preliminary data and a hypothesis foundation for future research into the role of the microbiome in the pathogenesis of APS-related pregnancies.

## Conclusion

In conclusion, pregnant women with antiphospholipid syndrome exhibit a distinct vaginal microbiome characterized by selective depletion of *Lactobacillus johnsonii* and enrichment of *Bifidobacterium dentium*. These microbial alterations could contribute to the increased risk of adverse pregnancy outcomes observed in APS. Vaginal microbial signatures therefore hold promise as noninvasive biomarkers for risk stratification and as potential targets for microbiome-based therapeutic strategies. Future longitudinal studies integrating immunologic and microbial profiling are recommended while considering host-microbiome interactions.

## ACKNOWLEDGMENTS

The authors thank all the pregnant women who generously participated in this study and provided valuable samples, all staff who collected clinical case data for this study, and colleagues who guided ethical applications.

This work was supported by the National Natural Science Foundation of China (82403347).

F.Y. conceived and designed the study, performed data analysis, and drafted the manuscript. N.L. participated in sample collection, data analysis and manuscript review. P.S. contributed to sample collection. L.W., H.Z., T.W., Y.J., and C.Y. participated in investigation and data curation. W.J. and W.S. provided clinical guidance. W.Z. contributed to funding. F.L., W.Z., and D.W., as corresponding authors, supervised the study and critically revised the manuscript.

All participants provided written informed consent for the publication of the data and results obtained. All authors consent to the publication of this manuscript.

## AUTHOR AFFILIATIONS

[1]Department of Obstetrics and Gynecology, Tongji Hospital, Tongji Medical College, Huazhong University of Science and Technology, Wuhan, Hubei, China
[2]National Clinical Research Center for Obstetrics and Gynecology, Tongji Hospital, Tongji Medical College, Huazhong University of Science and Technology, Wuhan, Hubei, China

## AUTHOR ORCIDs

Nary Long  http://orcid.org/0009-0004-0694-5483
Ling Feng  http://orcid.org/0000-0001-8833-5574
Zizhuo Wang  http://orcid.org/0000-0002-9659-0644
Wencheng Ding  http://orcid.org/0009-0004-0881-775X

## FUNDING

| Funder | Grant(s) | Author(s) |
| --- | --- | --- |
| National Natural Science Foundation of China | 82403347 | Zizhuo Wang |

## AUTHOR CONTRIBUTIONS

Yilin Fu, Conceptualization, Formal analysis, Writing – original draft | Nary Long, Formal analysis, Writing – original draft | Phannaroat Sourn, Resources | Weikun Li, Data curation, Investigation | Zhenzhen He, Data curation, Investigation | Weidong Tan, Data curation, Investigation | Junjie Yuan, Data curation, Investigation | Yuxin Chen, Data curation, Investigation | Jianli Wu, Supervision | Shaoshuai Wang, Supervision | Ling

Feng, Supervision, Writing – review and editing | Zizhuo Wang, Funding acquisition, Supervision, Writing – review and editing | Wencheng Ding, Supervision, Writing – review and editing

## DATA AVAILABILITY

The raw sequence data reported in this paper have been deposited in the Genome Sequence Archive (Genomics, Proteomics & Bioinformatics 2025) in National Genomics Data Center (Nucleic Acids Res 2025), China National Center for Bioinformation / Beijing Institute of Genomics, Chinese Academy of Sciences (GSA: CRA031013) that are publicly accessible at https://ngdc.cncb.ac.cn/gsa.

## ETHICS APPROVAL

The study was approved by the Medical Ethics Committee of Tongji Hospital, Tongji Medical College (TJ-IRB202412216), in accordance with the Declaration of Helsinki. Written informed consent was obtained from all participants prior to enrollment.

## ADDITIONAL FILES

The following material is available online.

### Supplemental Material

**Supplemental figure legends (Spectrum03882-25-s0001.docx).** Legends for Figures S1 to S4.
**Supplemental figures (Spectrum03882-25-s0002.pdf).** Figures S1 to S4.

### Open Peer Review

**PEER REVIEW HISTORY (review-history.pdf).** An accounting of the reviewer comments and feedback.

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
