## [Reviewer comments · Microbiology Spectrum]

Microbiology Spectrum

Distinct vaginal microbial signatures in pregnancies complicated by antiphospholipid syndrome: Depletion of *Lactobacillus johnsonii* and enrichment of *Bifidobacterium dentium*

Yilin Fu, Nary Long, Phannaroat Sourn, Weikun Li, Zhenzhen He, Weidong Tan, Junjie Yuan, Yuxin Chen, Jianli Wu, Shaoshuai Wang, Ling Feng, Zizhuo Wang, and Wencheng Ding

Corresponding Author(s): Wencheng Ding, Tongji Hospital of Tongji Medical College of Huazhong University of Science and Technology Library

Review Timeline:

Submission Date:	December 5, 2025
Editorial Decision:	January 12, 2026
Revision Received:	February 3, 2026
Accepted:	February 19, 2026

Editor: Wei-Hua Chen

Reviewer(s): Disclosure of reviewer identity is with reference to reviewer comments included in decision letter(s). The following individuals involved in review of your submission have agreed to reveal their identity: Darina Cejkova (Reviewer #1)

Transaction Report:

DOI: <https://doi.org/10.1128/spectrum.03882-25>

Re: Spectrum03882-25 (**Distinct vaginal microbial signatures in pregnancies complicated by antiphospholipid syndrome: Depletion of *Lactobacillus johnsonii* and enrichment of *Bifidobacterium dentium***)

Dear Prof. Wencheng Ding:

Thank you for the privilege of reviewing your work. Below you will find my comments, instructions from the Spectrum editorial office, and the reviewer comments.

Revision Guidelines

Sincerely,
Wei-Hua Chen
Editor
Microbiology Spectrum

Reviewer #2 (Comments for the Author):

"Distinct vaginal microbial signatures in pregnancies complicated by antiphospholipid syndrome..." by Lu and colleagues describes microbiome alterations associated with antiphospholipid syndrome and pregnancy.

Overall, this is a fairly straight-forward microbiome study. The used of full-length 16S at a depth of over 20,000 reads per sample makes this a really interesting dataset since many 16S studies use short amplicons, limiting species resolution.

That said, I have a major issue with the apparent frequency of APS in this study. Everything I've read (but I am not an expert) suggests that APS is a rare condition without widely accepted diagnostic criteria. This advocacy website cites a large 2019 study that suggests something like 1 in 2000 people are affected (<https://aps-support.org.uk/about-aps/how-common-is-it>). Given that, finding 33 APS patients out of a total sample of 421 (before any screening) corresponds to a >8% rate and 27% (33/123) in the final Fig 1 stratification. That is several orders of magnitude too high and suggests weak/incorrect diagnostic criteria. The introduction needs to cite relevant literature on the rareness of this condition and explain the observed frequency in this study relative to that. If the the patient categorization isn't correct, the entire analysis is called into question. To this point, the methods need to include more than just "study group comprised women diagnosed with APS". Based on what criteria?

Minor:

Both ordinations show subgroupings of healthy controls and a generally triangular pattern that has been shown previously by other labs as different "ecotypes" (e.g., Ravel lab). This isn't discussed. Honestly, the intro could use some key references and reviews on the current state of the art when describing the vaginal microbiome (around Line 91).

Not really minor, all bacterial species names need to be italicized. I'm not sure how this didn't get fixed before review

Line 111, define GBS first use

Line 163, please describe the tools and databases used to taxonomically assign the 16S sequences

Line 215, "Venn Diagram revealed high similarity..." A Venn diagram isn't really a type of analysis? You should be calculating Beta diversity (something like the Bray-Curtis measure used in the PCoA) and showing the range of dissimilarities (maybe a boxplot?). The Venn diagrams are OK as a supplement but don't use them as a type of analysis in the text.

Line 378 Date should be Data

Fig 2A - Y axis should be Shannon diversity (or something) not "OTU level"

Reviewer #3 (Comments for the Author):

In this study the authors characterize the vaginal microbiota of pregnant women and make comparisons to patients with and without obstetric anti phospholipid syndrome (APS). They examine specific microbiota features as they correlate with clinical variables and develop a logistic regression model based on microbiome and clinical features to predict APS. I found the manuscript to be easy to read and interpret and found the analysis strategy to be generally sound. One major methodologic concern is that not all women had APS positive screens as part of study inclusion, some were included simply because it was listed as part of medical history, if the authors could show some comparisons to those who had positive serologic testing at enrollment, and those who had it listed as part of medical history, this could address this concern. The authors also don't examine what could be very interesting variables about the pregnancies that were included in the study. As far as I can tell, the only variable examined is birth weight and gestational age at delivery. Was there any correlation between any other pregnancy complications such as abruptions, intra-uterine growth restriction, ect? The pregnancy outcomes among study participants at the time that microbiota are sampled is probably the most impacted by the microbiota described here.

Please see some other minor considerations below

Please place species/genus names in italics as this is convention to convey this information.

Page 5 lines 120-122, please specify if the collected lab values are from the mother or the infant

Page 8-9 lines 175-177 It is stated that normally distrusted variables were assessed with t-tests, were tests performed to ensure normality?

Page 15 lines 337-339. The authors did not show that complications arise from microbiota disruption. Given the observational nature of the study, its impossible to know if microbiota are causing the observed differences in this correlational analysis or are simply associated with them.

Limitations of the use of 16s rRNA sequencing for microbiota characterization need to be discussed in strengths and limitations section.

Figure 4. Was correlational analysis controlled for multiple comparisons? As far as I can tell by the figure there is a 15x15 table, which is a significant number of comparisons, even if the authors did not examine all detected OTUs in a similar way. The heat map table also requires a legend for all the abbreviations shown on the x axis of this tble.

In this study the authors characterize the vaginal microbiota of pregnant women and make comparisons to patients with and without obstetric anti phospholipid syndrome (APS). They examine specific microbiota features as they correlate with clinical variables and develop a logistic regression model based on microbiome and clinical features to predict APS. I found the manuscript to be easy to read and interpret and found the analysis strategy to be generally sound. One major methodologic concern is that not all women had APS positive screens as part of study inclusion, some were included simply because it was listed as part of medical history, if the authors could show some comparisons to those who had positive serologic testing at enrollment, and those who had it listed as part of medical history, this could address this concern. The authors also don't examine what could be very interesting variables about the pregnancies that were included in the study. As far as I can tell, the only variable examined is birth weight and gestational age at deliver. Was there any correlation between any other pregnancy complications such as abruptions, intra-uterine growth restriction, ect? The pregnancy outcomes among study participants at the time that microbiota are sampled is probably the most impacted by the microbiota described here.

Please see some other minor considerations below

Please place species/genus names in italics as this is convention to convey this information.

Page 5 lines 120-122, please specify if the collected lab values are from the mother or the infant

Page 8-9 lines 175-177 It is stated that normally distributed variables were assessed with t-tests, were tests performed to ensure normality?

Page 15 lines 337-339. The authors did not show that complications arise from microbiota disruption. Given the observational nature of the study, its impossible to know if microbiota are causing the observed differences in this correlational analysis.

Limitations of the use of 16s rRNA sequencing for microbiota characterization need to be discussed in strengths and limitations section.

Figure 4. Was correlational analysis controlled for multiple comparisons? As far as I can tell by the figure there is a 15x15 table, which is a significant number of comparisons, even if the authors did not examine all detected OTUs in a similar way. The heat map table also requires a legend for all the abbreviations shown on the x axis of this table.

Response to Reviewers

Dear reviewers,

Thank you for the careful and insightful evaluation of our manuscript. We have carefully considered the comment and made revisions to address the concerns raised. In response to the reviewers' comments, we have thoroughly revised the manuscript. The major revisions include: (i) clarification of the case–control study design and detailed description of APS diagnostic criteria based on the revised Sydney classification; (ii) expansion and refinement of the background on vaginal microbiome community state types and unsupervised clustering analyses; (iii) restructuring of β -diversity analyses with appropriate use of Bray–Curtis and Canberra distances; (iv) comprehensive revision of the correlation analyses with false discovery rate (FDR) correction for multiple comparisons; (v) expansion of clinical outcome variables and clarification of laboratory data sources; and (vi) correction of formatting, terminology, and methodological details throughout the manuscript.

We are grateful for the reviewers' thoughtful feedback and believe that the revised manuscript now more accurately and rigorously presents our findings. We sincerely hope that the revisions meet your expectations and look forward to your feedback. Below is a detailed response to the comment. The comment is presented in blue, and revisions are presented in red.

Yours sincerely,

Wencheng Ding

03/02/2026

Reviewer #2:

Comment 1# That said, I have a major issue with the apparent frequency of APS in this study. Everything I've read (but I am not an expert) suggests that APS is a rare condition without widely accepted diagnostic criteria. This advocacy website cites a large 2019 study that suggests something like 1 in 2000 people are affected (<https://aps-support.org.uk/about-aps/how-common-is-it>). Given that, finding 33 APS patients out of a total sample of 421 (before any screening) corresponds to a >8% rate and 27% (33/123) in the final Fig 1 stratification. That is several orders of magnitude too high and suggests weak/incorrect diagnostic criteria. The introduction needs to cite relevant literature on the rareness of this condition and explain the observed frequency in this study relative to that. If the the patient categorization isn't correct, the entire analysis is called into question. To this point, the methods need to include more than just "study group comprised women diagnosed with APS". Based on what criteria?

Response to comment 1:

Thank you for your valuable feedback regarding the proportion of patients with APS and the diagnostic criteria in this study. Your concerns were insightful, and they prompted us to carefully reconsider the unclear points in our original manuscript, which we have now addressed through specific revisions. We believe that with the clarifications provided below and the corresponding manuscript updates, your concerns will be thoroughly resolved. The major revisions and substantive improvements are detailed below.

1) Regarding the high proportion of APS patients in the study. This issue stems from a lack of clear description of the study design and patient source, which may have led to the misconception that this is a cohort study based on a general obstetric population. In fact, this is a case-control study, and all APS cases were sourced from our hospital's recurrent pregnancy loss specialty clinic, which is the largest high-risk pregnancy referral center in central China, covering multiple provinces. As is well known, APS is indeed rare among the general pregnant population (with a prevalence typically below 1%) (Duarte-García A, et al. 2019), but its prevalence significantly increases to 5%-20% in those with recurrent pregnancy loss (Galli M, et al. 2003; Mercier M, et al. 2024). This aligns with the information you referenced from the advocacy website, stating that " 1 in 6 recurrent miscarriages. " Therefore, the recruitment of cases from this high-risk clinic, along with the intentional case-control ratio set to ensure statistical power, resulted in 33 APS patients in the final cohort, which is entirely consistent with expectations. This does not reflect the prevalence in the general population but rather the study's design, which specifically targets high-risk patients with recurrent miscarriage. To avoid any misunderstanding, we have now clarified the study design and patient source in both the introduction and methodology sections, highlighting that this is a case-control study.

2) Regarding the diagnostic criteria. We completely agree that providing more detail on the diagnostic criteria is essential. In the revised methodology section, we have explicitly stated that all APS patients included in the study were diagnosed strictly according to the internationally recognized revised Sapporo (Sydney)

classification criteria (Miyakis S, et al. 2006). We have provided a detailed description of the required clinical criteria (including recurrent unexplained miscarriage before 10 gestational weeks or unexplained fetal death at ≥ 10 gestational weeks) and laboratory criteria (including persistent positivity for lupus anticoagulant, anticardiolipin antibodies (IgG/IgM), or anti- β_2 -glycoprotein I antibodies (IgG/IgM). Additionally, we emphasized that all diagnoses were confirmed by rheumatologists or maternal-fetal medicine specialists. This addition ensures the transparency and rigor of the diagnostic process and clarifies the uniform criteria for patient inclusion.

Below, we enumerate to revisions related to this comment:

1) Original text (Lines 67-71):

“Antiphospholipid syndrome (APS) is a systemic autoimmune disease defined by the persistent presence of antiphospholipid antibodies (aPL), and clinically manifests with thrombosis and pregnancy morbidity¹. Clinically, obstetric APS (OAPS) contributes substantially to recurrent pregnancy loss, fetal death, intrauterine growth restriction, and preeclampsia, posing major threats to maternal and fetal health².”

1) Revised text (Lines 63-72):

“Antiphospholipid syndrome (APS) is a **rare** systemic autoimmune disease , **with an estimated prevalence of approximately 0.05% in the general population**, defined by the persistent presence of antiphospholipid antibodies (aPL), and clinically manifests with thrombosis and pregnancy morbidity^{1,2}. Clinically, obstetric APS (OAPS) contributes substantially to recurrent pregnancy loss, fetal death, intrauterine growth restriction, and preeclampsia, posing major threats to maternal and fetal health³. **In**

clinical practice, APS is a leading cause of recurrent pregnancy loss (RPL), with studies indicating that APS-related pregnancy complications occur in approximately 15–20% of women with a history of multiple miscarriages, often complicating the management of such patients^{4,5}.”

2) Original text (Lines 105-107):

“This observational study was conducted at Tongji Hospital, Tongji Medical College, Huazhong University of Science and Technology from March 2024 to March 2025.”

2) Revised text (Lines 109-112):

“This **case-control** study was conducted at Tongji Hospital, Tongji Medical College, Huazhong University of Science and Technology (**a major tertiary referral center**) from March 2024 to March 2025. **We aimed to recruit APS group and controls (healthy pregnant women without APS) matched for gestational age at sampling.**”

3) Original text (Lines 114-116):

“All participants were managed by two senior obstetricians with more than 20 years of clinical experience, and all patients received standard pharmacological management”

3) Revised text (Lines 118-131):

“**All patients in the APS group were diagnosed according to the revised Sapporo (Sydney) classification criteria¹⁹. Diagnosis required the fulfillment of at least one clinical criterion (including recurrent unexplained miscarriage before 10 gestational weeks or unexplained fetal death at ≥ 10 gestational weeks) and one laboratory**

criterion, defined as persistent positivity for lupus anticoagulant, anticardiolipin antibodies (IgG/IgM), or anti- β_2 -glycoprotein I antibodies (IgG/IgM) on at least two occasions ≥ 12 weeks apart. All diagnoses were confirmed by rheumatologists or maternal-fetal medicine specialists. Among the APS group, some patients had been diagnosed with APS within 2 years prior to pregnancy, while others were newly identified during the current pregnancy; however, all cases fulfilled the revised Sydney classification criteria. All participants were under the care of two senior obstetricians with more than 20 years of clinical experience and received standard pharmacological management²⁰.”

Your feedback has helped us address critical gaps in the manuscript, for which we are deeply grateful. We have incorporated these clarifications into the introduction and methodology sections to better reflect the study's design and execution. We hope these revisions adequately address your concerns.

Comment 2# Both ordinations show subgroupings of healthy controls and a generally triangular pattern that has been shown previously by other labs as different "ecotypes" (e.g., Ravel lab). This isn't discussed. Honestly, the intro could use some key references and reviews on the current state of the art when describing the vaginal microbiome (around Line 91).

Response to comment 2:

We sincerely thank the reviewer for the insightful comments regarding the apparent substructures observed in the ordination plots, as well as for highlighting the

seminal work on vaginal community state types (CSTs). These valuable suggestions prompted us to substantially strengthen the background of our study and, more importantly, to perform deeper unsupervised analyses of our data. This additional work led to a meaningful finding that clarifies our conclusions and markedly reinforces the overall rigor of the manuscript. The major revisions and substantive improvements are detailed below.

1) As suggested, we expanded the Introduction to include key foundational studies on vaginal CSTs (e.g., Ravel et al., PNAS, 2011), thereby providing a more comprehensive and solid conceptual framework for the present study.

2) We are grateful for the reviewer's keen observation regarding potential ecotypes. We would first like to clarify that the principal coordinates analysis (PCoA) shown in the original submission was based on Canberra distance, a metric that is particularly sensitive to variation in low-abundance (rare) taxa. In contrast, the classical definition of vaginal CSTs is based on Bray-Curtis dissimilarity, which is more strongly driven by dominant taxa. Therefore, although the observed substructure in our ordination visually resembles inter-individual variation patterns, it more likely reflects differences in the vaginal "rare biosphere" rather than the canonical *Lactobacillus*-dominated CSTs. Consequently, despite the triangular pattern, these groupings may not fully correspond to the CSTs originally described by Ravel et al.

To further and systematically investigate this substructure, we performed hierarchical clustering using the same Canberra distance matrix and identified three stable clusters, which we termed Canberra types (CTs) to distinguish them from

classical CSTs (new Figure 3A). Notably, one cluster (CT2) contained 31 of the 33 APS samples, indicating a highly homogeneous and disease-associated vaginal microbiome state. In contrast, healthy control samples were distributed across the other two CTs (CT1 and CT3), reflecting natural inter-individual variability in low-abundance vaginal taxa during late pregnancy.

Taxonomic composition (new Figure 3B) and differential abundance analyses (new Figure 3C) demonstrated that CT2 is characterized by enrichment of *Bifidobacterium dentium* and depletion of *Lactobacillus johnsonii*, while the dominant taxa (*L. crispatus* and *L. iners*) showed no marked differences. These results are fully consistent with our earlier analyses in Figure 2, supporting the conclusion that APS is not associated with a gross shift in overall community structure, but rather with a distinct, low-abundance-driven compositional signature. Importantly, this analysis also provides a logical transition to the subsequent species-level investigations.

3) Based on these results, we added a new paragraph following the subsection “Vaginal Microbial Diversity” to describe this clustering analysis and its biological implications.

4) In the Discussion, we further elaborate on the significance of this finding. The presence of CT1 and CT3 among healthy pregnancies supports the notion that the tight and homogeneous clustering of APS samples into CT2 is unlikely to be random. Instead, it strongly suggests that APS-specific pathophysiological mechanisms may actively shape this distinct vaginal microbiome configuration.

Once again, we thank the reviewer for these constructive comments. The discussion of disease-related subgroups serves as an effective conceptual bridge between group-level differences in community structure and species-level compositional changes, substantially strengthening both the narrative coherence and the analytical depth of the manuscript.

Below, we enumerate to revisions related to this comment:

1) Original text (Lines 87-93):

“Beyond the gut, the vaginal microbiome is another pivotal ecosystem, represents a critical interface between mucosal immunity and pregnancy outcomes. A *Lactobacillus*-dominated microbiome, especially with *L. crispatus*, which maintain acidic pH, inhibit pathogens, and regulate local inflammation defines a healthy vaginal state. Conversely, dysbiosis characterized by reduced lactobacilli and overgrowth of anaerobes are associated with preterm birth, preeclampsia, and recurrent miscarriage¹¹⁻¹³. These conditions overlap with OAPS manifestations”

1) Revised text (Lines 87-97):

“Beyond the gut, the vaginal microbiome is another pivotal ecosystem, **representing** a critical interface between mucosal immunity and pregnancy outcomes. **In reproductive-age women individuals, the vaginal microbiome is often categorized into five community state types (CSTs) based on the dominant bacterial species¹⁴.** **During pregnancy, the microbiome generally stabilizes towards a *Lactobacillus*-dominated state. A *L. crispatus*-dominated microbiome is particularly associated with health due to its role in maintaining** acidic pH, **inhibiting** pathogens,

and regulating local inflammation¹⁵. Conversely, dysbiosis characterized by reduced lactobacilli and overgrowth of anaerobes is associated with adverse outcomes such as preterm birth, preeclampsia, and recurrent miscarriage¹⁶⁻¹⁸. These conditions overlap with OAPS manifestations ”

2) Revised text (Lines 262-274): **Additional content in the Results section**

“Clustering analysis reveals a disease-associated vaginal microbiome type

To further decipher the community structure captured by the Canberra distance, we performed hierarchical clustering on the species-level abundance data. This analysis delineated three primary clusters, which we designated Canberra Type 1 (CT1), CT2, and CT3 (Figure 3A). Strikingly, the APS group samples were almost exclusively concentrated within CT2 (31 of 33 APS samples), while healthy controls were distributed across CT1 and CT3. This indicates a high degree of microbial homogeneity among APS pregnancies that is distinct from the variation observed in healthy pregnancies. Visualization of the taxonomic composition (Figure 3B) and identification of the most differentially abundant species (Figure 3C) within these clusters confirmed that CT2 was characterized by the unique vaginal microbiome pattern: significant enrichment of *Bifidobacterium dentium* and depletion of *Lactobacillus johnsonii*.”

3) Revised text (Lines 344-356): **Additional content in the Discussion section**

“Our unsupervised clustering analysis not only identified a distinct and highly homogeneous vaginal microbiome state (CT2) encompassing the majority of pregnancies complicated by antiphospholipid syndrome (APS), but also revealed that

the healthy vaginal microbiome in late gestation could be further stratified into at least two additional community types (CT1 and CT3), primarily distinguished by variation in low-abundance taxa. Although the ecological drivers and structural characteristics underlying this substructure within the healthy state remain to be elucidated and were beyond the scope of the present study, its presence nonetheless supports the non-random nature of the tight clustering of APS samples into a CT2 vaginal community characterized by enrichment of *Bifidobacterium dentium* and depletion of *Lactobacillus johnsonii*. Collectively, these findings suggest that this microbiome configuration is strongly linked to and may be a consequence of APS-specific pathophysiological mechanisms.”

Comment 3# Not really minor, all bacterial species names need to be italicized. I'm not sure how this didn't get fixed before review.

Response to comment 3:

Thank you for pointing this out. We apologize for this oversight. All bacterial species names have now been carefully checked and consistently italicized throughout the revised manuscript.

Comment 4# Line 111, define GBS first use.

Response to comment 4:

Thank you for your valuable suggestion. We have now defined "GBS" as "Group B Streptococcus" upon its first use in the manuscript.

Comment 5# Line 215, "Venn Diagram revealed high similarity..." A Venn diagram isn't really a type of analysis? You should be calculating Beta diversity (something like the Bray-Curtis measure used in the PCoA) and showing the range of dissimilarities (maybe a boxplot?). The Venn diagrams are OK as a supplement but don't use them as a type of analysis in the text.

Response to comment 5:

We sincerely thank the reviewer for these insightful and highly constructive comments. Your suggestions that Venn diagrams should not be considered an analytical approach and that β -diversity should be assessed and the range of dissimilarity should be explicitly presented by boxplot were particularly valuable, and they provided clear guidance for substantially improving both the analytical rigor and the interpretability of our results. We fully agree with these points and have revised the manuscript accordingly. The major revisions and substantive improvements are detailed below.

1) Introduction of β -diversity distance distribution boxplots to restructure the analytical framework. We fully acknowledge that Venn diagrams merely describe overlaps in OTU presence/absence and do not quantify the degree of community dissimilarity. In response to your recommendation, we therefore calculated β -diversity based on Bray–Curtis dissimilarity and quantified within-group variability. Specifically, pairwise Bray–Curtis distances were calculated among all samples within each group, and the distributions of these distances were visualized using boxplots, which are now presented as the main result in Figure 2B. This analysis

showed that although the APS group exhibited a trend toward lower within-group dissimilarity (median 0.77, IQR: 0.11–0.84) compared with the control group (median 0.78, IQR: 0.29–0.84), the difference did not reach statistical significance ($P = 0.087$). These results indicate that overall community variability between APS and control groups is broadly comparable, with no pronounced separation at the level of dominant taxa.

2) Repositioning the role of the Venn diagram as descriptive supplementary information. We fully agree with your assessment regarding the appropriate role of Venn diagrams. In the revised manuscript, the Venn diagram is no longer described as an independent “analysis,” and all related statements in the main text have been removed. The figure has been relocated to the Supplementary Materials (Supplementary Figure). In the main text, the Venn diagram is referenced only descriptively, in support of the Bray–Curtis distance results shown in Figure 2B. It is used solely to illustrate a trend toward reduced compositional heterogeneity in the APS vaginal microbiota, reflected by a lower number of observed taxa compared with controls, rather than as evidence of community dissimilarity.

3) Refinement of the rationale for using Canberra distance. Following your guidance, we first interpreted the Bray–Curtis–based β -diversity results to show that the dominant taxa and overall community structure do not differ markedly between groups. However, the observed trend toward reduced heterogeneity in the APS group led us to hypothesize that intergroup differences may instead be driven by low-abundance or rare taxa. Accordingly, we selected the Canberra distance—known

to be more sensitive to changes in rare taxa—as the primary metric for intergroup β -diversity comparisons (Figure 2C, D). Using this distance, clear group separation was observed ($R = 0.074$, $P = 0.027$). Together, the sequential use of Bray-Curtis and Canberra distances provides a coherent and stepwise analytical rationale, jointly supporting our central conclusion that APS may be associated with selective community restructuring rather than broad shifts in dominant microbial composition.

4) Reorganization of sequencing quality control descriptions. Descriptions of rarefaction analysis and pan-core OTU analyses have been moved to the beginning of the microbiome diversity subsection, improving the logical flow and clarity of the Results section.

Once again, we greatly appreciate your constructive feedback. In response, we have thoroughly reorganized the “Vaginal microbial diversity” subsection of the Results, resulting in a clearer analytical structure and stronger alignment between methodology, results, and interpretation.

Below, we enumerate to revisions related to this comment:

1) Original text (Lines 165-168):

“Alpha diversity indices including Simpson, Shannon, and Chao1 richness were calculated with Mothur v1.30.1. Beta diversity was determined by principal coordinate analysis (PCoA) based on Bray-curtis dissimilarity using Vegan v2.5-3 package.”

1) Revised text (Lines 190-196):

“Alpha diversity was assessed using the Shannon index calculated with Mothur v1.30.1. Beta diversity was determined based on both Bray-Curtis and Canberra dissimilarity metrics, visualized via principal coordinate analysis (PCoA) and non-metric multidimensional scaling (NMDS). The statistical significance of group separation in beta diversity was tested using Analysis of Similarities (ANOSIM) determining statistical significance.”

2) Original text (Lines 201-217):

“Vaginal Microbial Diversity

Alpha diversity analysis (Shannon indices) showed no statistically significant differences (all $P > 0.05$) between the APS and control groups (Figure 2A), suggesting that overall diversity remained comparable between groups. In contrast, beta diversity analyses revealed significant microbial community compositional differences. NMDS of OTU abundances, using Canberra distances, revealed a moderate level of ordination fit. Despite this, PERMANOVA testing demonstrated statistically significant effects of group on beta diversity ($R^2=0.074$, $P=0.027$), as visualized by PCoA (Figure 2B-C). These findings suggest that while overall microbial diversity remains comparable between groups, APS pregnancies harbor distinct microbial community structures.

Vaginal microbiome differences between APS and control groups

Rarefaction and pan-core OTU analyses confirmed adequate sequencing depth and comprehensive coverage of vaginal microbiome diversity within the study population (Figure S1A-C). Venn Diagram revealed high similarity between groups,

suggesting comparable overall taxonomic diversity in phyla, genera, species and OTUs (Figure S2A-D).”

2) Revised text (Lines 240-261):

“Vaginal Microbial Diversity

Rarefaction and pan-core OTU analyses confirmed adequate sequencing depth and comprehensive coverage of vaginal microbiome diversity within the study population (Figure S2A-C). Alpha diversity analysis (Shannon indices) showed no statistically significant differences (all $P > 0.05$) between the APS and control groups (Figure 2A), suggesting the two groups had similar overall levels of species richness and evenness.

Using the Bray-Curtis distance, which is sensitive to overall community variation, we calculated the within-group dissimilarities for each group (Figure 2B). The APS group exhibited a trend of lower within-group heterogeneity (0.77, IQR: 0.11-0.84) compared to the control group (0.78, IQR: 0.29-0.84), although this difference did not reach the significance threshold ($P = 0.087$). This finding is consistent with the results from the Venn diagram, which showed that the APS group harbored fewer unique taxa at various taxonomic levels than the control group (Figure S3A-D). This suggests that the microbial composition in the APS group may be more homogeneous and possibly associated with a more consistent internal community state.

To further investigate subtle differences in the overall community structure between the two groups, we conducted analysis using the Canberra distance, which is more sensitive to changes in rare taxa. NMDS and PCoA of OTU abundances,

revealed a moderate level of ordination fit **and** demonstrated statistically significant effects of group on beta diversity ($R=0.074$, $P=0.027$) (Figure 2**C-D**). These findings suggest that while overall microbial diversity remains comparable between groups, APS pregnancies **may** harbor distinct microbial community structures.”

Comment 6# Line 163, please describe the tools and databases used to taxonomically assign the 16S sequences.

Response to comment 6:

We thank the reviewer for highlighting the need for additional methodological details. In the revised manuscript, we have explicitly specified the tools and databases used for taxonomic annotation, and we have carefully reviewed and streamlined the descriptions of the bioinformatics analyses to ensure clarity and completeness. Below, we enumerate to revisions related to this comment:

1) Original text (Lines 163-172):

“Bioinformatic analysis

Bioinformatic analysis of the microbiota was carried out using the Majorbio Cloud platform (<https://cloud.majorbio.com>). Alpha diversity indices including Simpson, Shannon, and Chao1 richness were calculated with Mothur v1.30.1. Beta diversity was determined by principal coordinate analysis (PCoA) based on Bray-curtis dissimilarity using Vegan v2.5-3 package. Differences in microbial community composition between groups were assessed using Wilcoxon rank-sum tests. The linear discriminant analysis (LDA) effect size (LEfSe) was performed to identify the

significantly abundant taxa (phylum to species) of bacteria among the different groups (LDA score > 2, P < 0.05) (<http://huttenhower.sph.harvard.edu/LEfSe>).”

1) Revised text (Lines 185-199):

“Bioinformatic analysis

Bioinformatic analysis was performed using the Majorbio Cloud platform (<https://cloud.majorbio.com>). Downstream statistical analyses were conducted based on the rarefied OTU table. The taxonomic classification of each OTU representative sequence was carried out using the RDP Classifier (version 2.2) against the 16S rRNA gene database (NT_Taxon_core_v2024) with a confidence threshold of 0.70. Alpha diversity was assessed using the Shannon index calculated with Mothur v1.30.1. Beta diversity was determined based on both Bray-Curtis and Canberra dissimilarity metrics, visualized via principal coordinate analysis (PCoA) and non-metric multidimensional scaling (NMDS). The statistical significance of group separation in beta diversity was tested using Analysis of Similarities (ANOSIM) determining statistical significance. Differences in microbial community composition between groups were assessed using Wilcoxon rank-sum tests. The linear discriminant analysis (LDA) effect size (LEfSe) was performed to identify the significantly abundant taxa (phylum to species) of bacteria among the different groups (LDA score > 2, P < 0.05).”

Comment 7# Line 378 Date should be Data.

Response to comment 7:

Thank you for pointing out this typo. We have corrected “Date” to “Data” in the manuscript.

Comment 8# Fig 2A - Y axis should be Shannon diversity (or something) not "OTU level".

Response to comment 8:

Thank you for this helpful suggestion. We have updated the Y-axis label in Figure 2A to “Shannon diversity” to accurately reflect the diversity metric used.

Reviewer #3:

Comment 1# One major methodologic concern is that not all women had APS positive screens as part of study inclusion, some were included simply because it was listed as part of medical history, if the authors could show some comparisons to those who had positive serologic testing at enrollment, and those who had it listed as part of medical history, this could address this concern.

Response to comment 1:

Thank you for raising this important methodological concern regarding the consistency of APS diagnosis within our study cohort. We fully agree that this issue requires explicit clarification and have addressed it through the following revisions and additional analyses:

1) In the revised Methods (Study design and population), we have added detailed information regarding the timing of APS diagnosis: “Among the APS group, some patients had been diagnosed with APS within 2 years prior to pregnancy, while others were newly identified during the current pregnancy; however, all cases fulfilled the revised Sydney classification criteria.” This clarification confirms that, regardless of the timing of diagnosis, all patients met the same strict and internationally accepted diagnostic criteria.

2) To further address the potential concern of diagnostic timing-related heterogeneity, we added a supplementary analysis in the Results (Participants and clinical characteristics). We first report the cohort composition: “Among the 33 women in the APS group, 19 had been diagnosed with APS prior to pregnancy, while

14 were newly diagnosed during the current pregnancy.” More importantly, we subsequently state that “ Notably, comparisons of microbial diversity and community composition between Pre-APS subgroup and those with Index-APS subgroup revealed no significant differences (Figure S1A-C).”

We sincerely appreciate this constructive comment. We believe that the concern regarding potential bias arising from differences in diagnostic timing has now been adequately addressed, and that this suggestion has helped us provide a more detailed and rigorous characterization of the study population.

Below, we enumerate to revisions related to this comment:

1) Original text (Lines 114-116):

“All participants were managed by two senior obstetricians with more than 20 years of clinical experience, and all patients received standard pharmacological management”

1) Revised text (Lines 118-131):

“All patients in the APS group were diagnosed according to the revised Sapporo (Sydney) classification criteria¹⁹. Diagnosis required the fulfillment of at least one clinical criterion (including recurrent unexplained miscarriage before 10 gestational weeks or unexplained fetal death at ≥ 10 gestational weeks) and one laboratory criterion, defined as persistent positivity for lupus anticoagulant, anticardiolipin antibodies (IgG/IgM), or anti- β_2 -glycoprotein I antibodies (IgG/IgM) on at least two occasions ≥ 12 weeks apart. All diagnoses were confirmed by rheumatologists or maternal-fetal medicine specialists. Among the APS group, some patients had been

diagnosed with APS within 2 years prior to pregnancy, while others were newly identified during the current pregnancy; however, all cases fulfilled the revised Sydney classification criteria. All participants were under the care of two senior obstetricians with more than 20 years of clinical experience and received standard pharmacological management²⁰.”

2) Original text (Lines 191-200):

“The final cohort comprised 123 participants, among these, 33 pregnant women were defined as the APS group based on APS-related antibody testing or their medical history, while 90 healthy pregnant women served as the control group, matched to the APS group for age and gestational age at sampling. Baseline clinical characteristics are summarized in Table 1. There were no significant differences in age and gestational age at sampling between the two groups. Compared with controls, the APS group had higher gravidity (2.73 ± 1.28 vs. 1.50 ± 0.78 , $P < 0.001$), more frequent spontaneous abortion (1.52 ± 1.09 vs. 0.38 ± 0.63 , $P < 0.001$), and a greater proportion of cesarean delivery (87.9% vs. 56.7%, $P = 0.001$)”

2) Revised text (Lines 224-238):

“The final cohort comprised 123 participants, including 33 pregnant women meeting the revised Sydney criteria for APS, and 90 matched healthy pregnant women. Of the 33 APS pregnancies, 19 had been diagnosed prior to pregnancy (Pre-APS) and 14 were newly diagnosed during the index pregnancy (Index-APS). Baseline clinical characteristics are summarized in Table 1. The two groups were matched for gestational age at sampling, as per design. Compared with controls, the

APS group was slightly older than the control group (median age: 33.0 vs. 31.0 years, $P=0.024$), and had higher gravidity (2.73 ± 1.28 vs. 1.50 ± 0.78 , $P<0.001$), more frequent spontaneous abortion (1.52 ± 1.09 vs. 0.38 ± 0.63 , $P<0.001$), and a greater proportion of cesarean delivery (87.9% vs. 56.7%, $P=0.001$). In terms of pregnancy outcomes, the APS group had a lower birth weight percentile (29.8% vs. 45.8%, $P=0.039$) and a higher incidence of placental abruption (12.1% vs. 2.2%, $P=0.044$). Notably, comparisons of microbial diversity and community composition between Pre-APS subgroup and those with Index-APS subgroup revealed no significant differences (Figure S1A-C).”

Comment 2# The authors also don't examine what could be very interesting variables about the pregnancies that were included in the study. As far as I can tell, the only variable examined is birth weight and gestational age at delivery. Was there any correlation between any other pregnancy complications such as abruptions, intra-uterine growth restriction, ect? The pregnancy outcomes among study participants at the time that microbiota are sampled is probably the most impacted by the microbiota described here.

Response to comment 2:

We sincerely thank the reviewer for this insightful and important suggestion. We fully agree that exploring the relationship between the vaginal microbiome and specific pregnancy complications is of great clinical relevance and is essential for

understanding its potential implications. The major revisions and substantive improvements are detailed below.

1) Following your guidance, we have systematically expanded the analysis of pregnancy outcomes in the revised manuscript. We additionally collected and analyzed detailed data on placental abruption, small for gestational age (SGA), large for gestational age (LGA), premature rupture of membranes (PROM), preterm birth, and birth weight percentiles. These results have been summarized in table (Table 1). Our analysis showed that the incidence of placental abruption was significantly higher in the APS pregnancy group than in healthy controls (12.1% vs. 2.2%, $P = 0.044$), and that birth weight percentiles were significantly lower in the APS group. These findings are consistent with the known pathophysiology of APS.

2) Regarding your suggestion to directly analyze associations between the vaginal microbiome and these specific complications, we carefully evaluated this possibility. The primary aim of the present study was to provide a first characterization of the vaginal microbiome in pregnant women with APS using a case-control design. However, the sample size of the APS group ($n = 33$) limits the statistical power for reliable within-group subgroup analyses (e.g., comparing microbiome features between APS patients with and without specific complications). For instance, only four APS patients in our cohort experienced placental abruption, and conducting statistical comparisons with such small numbers would carry a high risk of false-negative or false-positive findings, thereby compromising the robustness of any conclusions.

3) For these reasons, we believe that investigating these highly compelling “microbiome-outcome” associations would be more appropriately addressed in future, specifically designed, large-scale prospective APS cohorts. We have added this point to the Strengths and Limitations section of the manuscript. Nevertheless, the comprehensive outcome data now included in the revised manuscript provide a clear variable framework and preliminary clinical context for future studies.

Once again, we sincerely appreciate the reviewer’s valuable comments, which prompted us to enrich the clinical dimension of our study and to clearly define important directions for future research.

Below, we enumerate to revisions related to this comment:

1) Original text (Lines 117-122):

“Clinical data were retrospectively collected, including maternal age, gravidity, parity, gestational age at delivery, neonatal birth weight, admission to the neonatal intensive care unit (NICU), complete blood count indices including red blood cell count (RBC), platelet count (PLT), and coagulation parameters including antithrombin activity (AT), activated partial thromboplastin time (APTT), and thrombin time (TT).”

1) Revised text (Lines 132-144):

“Clinical data were retrospectively collected, **Maternal characteristics** included age, gravidity, parity. **Pregnancy and delivery outcomes** comprised gestational age at delivery, **mode of delivery** (categorized as cesarean section or vaginal delivery), and **the occurrence of specific complications: preterm birth (<37 weeks),**

small-for-gestational-age (SGA, birth weight <10th percentile), large-for-gestational-age (LGA, birth weight >90th percentile), premature rupture of membranes (PROM), and placental abruption. Neonatal outcomes included neonatal birth weight, birth weight percentile (calculated using the INTERGROWTH-21st standards), Apgar scores at 1 and 5 minutes, and admission to the neonatal intensive care unit (NICU). Maternal laboratory indices included complete blood count indices including red blood cell count (RBC), platelet count (PLT), and coagulation parameters including antithrombin activity (AT), activated partial thromboplastin time (APTT), and thrombin time (TT)”

2) Original text (Lines 196-200):

“Baseline clinical characteristics are summarized in Table 1. There were no significant differences in age and gestational age at sampling between the two groups. Compared with controls, the APS group had higher gravidity (2.73 ± 1.28 vs. 1.50 ± 0.78 , $P<0.001$), more frequent spontaneous abortion (1.52 ± 1.09 vs. 0.38 ± 0.63 , $P<0.001$), and a greater proportion of cesarean delivery (87.9% vs. 56.7%, $P=0.001$).”

2) Revised text (Lines 228-234):

“Baseline clinical characteristics are summarized in Table 1. The two groups were matched for gestational age at sampling, as per design. Compared with controls, the APS group was slightly older than the control group (median age: 33.0 vs. 31.0 years, $P=0.024$), and had higher gravidity (2.73 ± 1.28 vs. 1.50 ± 0.78 , $P<0.001$), more frequent spontaneous abortion (1.52 ± 1.09 vs. 0.38 ± 0.63 , $P<0.001$), and a greater proportion of cesarean delivery (87.9% vs. 56.7%, $P=0.001$). In terms of

pregnancy outcomes, the APS group had a lower birth weight percentile (29.8% vs. 45.8%, $P=0.039$) and a higher incidence of placental abruption (12.1% vs. 2.2%, $P=0.044$).”

3) Original text (Lines 353-355):

“Several limitations warrant acknowledgment. First, as a descriptive and exploratory study, the cross-sectional nature precludes causal inference between microbial alterations and APS-related outcomes.”

3) Revised text (Lines 423-427):

“Several limitations warrant acknowledgment. First, as a descriptive and exploratory study, the cross-sectional nature **and limited sample size, particularly within the APS subgroup**, precludes **robust assessment of associations** between **specific** microbial alterations and **individual pregnancy complications and preclude causal inference.**”

Comment 3# Please place species/genus names in italics as this is convention to convey this information.

Response to comment 3:

Thank you for your helpful suggestion. We have now italicized all species and genus names in the revised manuscript, in accordance with standard taxonomic conventions.

Comment 4# Page 5 lines 120-122, please specify if the collected lab values are from the mother or the infant.

Response to comment 4:

Thank you for your suggestion. We have clarified in the revised manuscript that the laboratory values for complete blood count (RBC, PLT) and coagulation parameters (AT, APTT, TT) were collected from the mother, as requested.

We enumerate to revisions related to this comment below:

1) Original text (Lines 119-122):

“complete blood count indices including red blood cell count (RBC), platelet count (PLT), and coagulation parameters including antithrombin activity (AT), activated partial thromboplastin time (APTT), and thrombin time (TT).”

1) Revised text (Lines 141-144):

“**Maternal laboratory indices included** complete blood count indices including red blood cell count (RBC), platelet count (PLT), and coagulation parameters including antithrombin activity (AT), activated partial thromboplastin time (APTT), and thrombin time (TT)”

Comment 5# Page 8-9 lines 175-177 It is stated that normally distrusted variables were assessed with t-tests, were tests performed to ensure normality?

Response to comment 5:

We thank the reviewer for pointing out the omission in our description of the statistical methods. To ensure the appropriateness of the statistical tests applied, we assessed the distribution of all continuous variables prior to analysis.

In the revised manuscript, we have added a clarification to the Statistical Analysis section stating that “normality of all continuous variables was assessed using the

Shapiro–Wilk test.” Accordingly, variables such as maternal age and gestational age at sampling, which showed non-normal distributions, are presented as medians with interquartile ranges and were analyzed using the Wilcoxon rank-sum test.

This revision ensures full transparency and methodological rigor of the statistical analyses. We appreciate the reviewer’s helpful comment, which allowed us to improve the clarity and completeness of this section.

Below, we enumerate to revisions related to this comment:

1) Original text (Lines 175-178):

“Continuous variables with normal distribution were analyzed using two-tailed Student's t-tests, while non-normally distributed data were assessed with Wilcoxon rank-sum tests.”

1) Revised text (Lines 202-205):

“**The normality of all continuous variables was assessed using the Shapiro-Wilk test. Based on this assessment,** continuous variables with normal distribution were analyzed using two-tailed Student's t-tests, while non-normally distributed data were assessed with Wilcoxon rank-sum tests. ”

Comment 6# Page 15 lines 337-339. The authors did not show that complications arise from microbiota disruption. Given the observational nature of the study, its impossible to know if microbiota are causing the observed differences in this correlational analysis or are simply associated with them.

Response to comment 6:

We sincerely thank the reviewer for this important and constructive comment. We fully agree that, given the observational and cross-sectional nature of the present study, our findings can demonstrate associations rather than causal relationships. To ensure that our interpretation strictly adheres to this principle, we have revised the relevant statements in the Discussion to adopt more rigorous and cautious wording.

We enumerate to revisions related to this comment as follows:

1) Original text (Lines 281-286):

“This species-specific dysbiosis suggests that APS exerts selective ecological pressures rather than inducing a global microbial dysbiosis. Similar observations have been reported in pregnancy complicated by type 1 diabetes, where immune dysregulation shapes microbial community structure without reducing overall diversity¹⁶. These parallels highlight that distinct maternal disease contexts may drive unique microbial signatures”

1) Revised text (Lines 338-343):

“This species-specific dysbiosis suggests that APS **is associated with a selective reshaping of the microbial ecology**, rather than inducing a global microbial dysbiosis. Similar observations have been reported in pregnancy complicated by type 1 diabetes, where immune dysregulation shapes microbial community structure without reducing overall diversity. These parallels highlight that distinct maternal disease contexts may **be associated with** unique microbial signatures”

2) Original text (Lines 305-306):

“Its depletion likely reduces microbial resilience and increases susceptibility to dysbiosis”

2) Revised text (Lines 375-376):

“Its depletion **is thus positioned as a potential factor that may** reduce microbial resilience and increases susceptibility to dysbiosis”

3) Original text (Lines 312-313):

“Consequently, this immune-mediated instability may favor the colonization of atypical or opportunistic taxa, such as *B. dentium*”

3) Revised text (Lines 383-384):

“Consequently, this immune-mediated instability may **be permissive for** the colonization of atypical or opportunistic taxa, such as *B. dentium*”

Comment 7# Limitations of the use of 16s rRNA sequencing for microbiota characterization need to be discussed in strengths and limitations section.

Response to comment 7:

We thank the reviewer for raising this important point. As suggested, we have now explicitly acknowledged the methodological limitation of 16S rRNA gene sequencing in the revised “Strengths and Limitations” section.

We enumerate to revisions related to this comment below:

1) Original text (Lines 361-368):

“Furthermore, this is a single-center study in which all participants were recruited from the same hospital over a one-year period. Therefore, the generalizability and

broader applicability of the findings need to be validated in future multicenter, large-sample prospective cohort studies. Nonetheless, this study provides the first systematic description of vaginal microbiome alternations in pregnant women with APS, offering important preliminary data and hypothesis foundation for future research into the role of the microbiome in the pathogenesis of APS-related pregnancies.”

1) Revised text (Lines 434-444):

“Furthermore, this is a single-center study in which all participants were recruited from the same hospital over a one-year period. **Finally, the use of 16S rRNA gene sequencing, while providing taxonomic resolution, precludes functional inferences.** Therefore, the generalizability and broader applicability of the findings need to be validated in future multicenter, large-sample prospective cohort studies, **ideally incorporating multi-omics approaches such as metagenomics and metabolomics to elucidate functional mechanisms.** Nonetheless, this study provides the first systematic description of vaginal microbiome alternations in pregnant women with APS, offering important preliminary data and hypothesis foundation for future research into the role of the microbiome in the pathogenesis of APS-related pregnancies.”

Comment 8# Figure 4. Was correlational analysis controlled for multiple comparisons? As far as I can tell by the figure there is a 15x15 table, which is a significant number of comparisons, even if the authors did not examine all detected OTUs in a similar way. The heat map table also requires a legend for all the

abbreviations shown on the x axis of this tble.

Response to comment 8:

1) Revised text (Lines 207-210): **Additional content in the Methods section**

“ Spearman’s rank correlation analysis was conducted to assess relationships between vaginal microbiota and clinical parameters. Multiple testing was adjusted using the Benjamini-Hochberg FDR method within each analysis tier, with $q < 0.05$ considered significant.”

2) Original text (Lines 238-253):

“Association Between Vaginal Microbiome and Clinical Characteristics

Correlation heatmap analysis revealed opposing association patterns between the two key microbial biomarkers (*B. dentium* and *L. johnsonii*) and clinical characteristics (Figure 4A). Regarding hematological parameters (Figures 4B-C), *B. dentium* abundance was negatively correlated with both RBC count ($r=-0.220$, $P=0.016$) and PLT ($r=-0.190$, $P=0.038$), whereas *L. johnsonii* abundance showed a positive correlation with RBC count ($r=0.237$, $P=0.009$). For coagulation indicators, *L. johnsonii* abundance was negatively associated with APTT ($r=-0.186$, $P=0.044$) (Figure S3B). In terms of pregnancy-related variables, *B. dentium* correlated positively with gravidity ($r=0.226$, $P=0.012$) (Figure S3C) and spontaneous abortion history ($r=0.225$, $P=0.012$) (Figure 4D), but negatively correlated with gestational age at delivery ($r=-0.238$, $P=0.010$) (Figures 4E). In contrast, *L. johnsonii* displayed opposing association with gravidity ($r=-0.247$, $P=0.006$) (Figure S3D), spontaneous abortion history ($r=-0.225$, $P=0.004$) (Figure 4D), and gestational age at delivery

($r=0.240$, $P=0.009$) (Figures 4E). These associations link microbial composition to both hematologic and obstetric outcomes in APS.”

2) Revised text (Lines 294-308):

“Association Between Vaginal Microbiome and Clinical Characteristics

Exploratory Spearman correlation analysis was performed to assess associations between vaginal taxa and clinical parameters. After applying FDR correction for multiple comparisons, none of the correlations remained statistically significant (all $q > 0.05$). The patterns of nominal (uncorrected) associations are displayed in a heatmap (Figure 5A). Regarding hematological parameters, *B. dentium* abundance showed nominal negative correlations with both RBC count ($r=-0.220$, $P=0.016$) and PLT ($r=-0.190$, $P=0.038$), whereas *L. johnsonii* abundance showed a nominal positive correlation with RBC count ($r=0.237$, $P=0.009$). In terms of pregnancy-related variables, *B. dentium* showed a nominal positive correlation with gravidity ($r=0.226$, $P=0.012$) and spontaneous abortion history ($r=0.225$, $P=0.012$), but a nominal negative correlation with gestational age at delivery ($r=-0.238$, $P=0.010$). *L. johnsonii* displayed opposing nominal association patterns with these same variables. These exploratory findings suggest potential links between the vaginal microbiome and clinical features in APS that warrant validation in larger, dedicated cohorts.”

3) Original text (Lines 552-560):

“Figure 4. Associations between the vaginal microbiome and clinical characteristics

(A)Heatmap showing correlations between vaginal microbial species and clinical

parameters, calculated using Spearman's rank correlation. (B-C) Scatter plots illustrating associations of *Bifidobacterium dentium* and *Lactobacillus johnsonii* with hematological indices, including RBC count and platelet count, assessed by Spearman's correlation. (D-E) Scatter plots illustrating associations of *Bifidobacterium dentium* and *Lactobacillus johnsonii* with spontaneous abortion and gestational age at delivery, assessed by Spearman's correlation. ”

3) Revised text (Lines 661-673):

“Figure 5. Associations between the vaginal microbiome and clinical characteristics

(A)Heatmap showing correlations between vaginal microbial species and clinical parameters, calculated using Spearman's rank correlation. **P-values were adjusted for multiple comparisons using the Benjamini-Hochberg false discovery rate (FDR) procedure. Asterisks within cells denote the significance level of the FDR-adjusted q-value: *q < 0.05, **q < 0.01, ***q < 0.001. The color gradient represents the strength and direction of correlation, with red indicating positive correlations and blue indicating negative correlations. Abbreviations: RBC, red blood cell count; HB, hemoglobin; PLT = platelet count; PT, prothrombin time; APTT, activated partial thromboplastin time; TT, thrombin time; FIB, fibrinogen; G, gravidity; P, parity; SA, spontaneous abortion; IA, induced abortion; GAD, gestational age at delivery; BW, birth weight; C-section, cesarean section.”**

Re: Spectrum03882-25R1 (**Distinct vaginal microbial signatures in pregnancies complicated by antiphospholipid syndrome: Depletion of Lactobacillus johnsonii and enrichment of Bifidobacterium dentium**)

Dear Prof. Wencheng Ding:

Congratulations!

Your manuscript has been accepted, and I am forwarding it to the ASM production staff for publication. Your paper will first be checked to make sure all elements meet the technical requirements. ASM staff will contact you if anything needs to be revised before copyediting and production can begin. Otherwise, you will be notified when your proofs are ready to be viewed.

Sincerely,
Wei-Hua Chen
Editor
Microbiology Spectrum

Reviewer #2 (Comments for the Author):

The authors have addressed my previous concerns about study design and vaginal ecotypes.

Reviewer #3 (Comments for the Author):

My concerns have been addressed

Reviewer #4 (Comments for the Author):

Dear Authors,

Thanks for addressing all the concerns raised in the initial review.